# Warpspeed Computation of Optimal Transport, Graph Distances, and Embedding Alignment

## Abstract

Optimal transport (OT) is a cornerstone of many machine learning tasks. The current best practice for computing OT is via entropy regularization and Sinkhorn iterations. This algorithm runs in quadratic time and requires calculating the full pairwise cost matrix, which is prohibitively expensive for large sets of objects. To alleviate this limitation we propose to instead use a sparse approximation of the cost matrix based on locality sensitive hashing (LSH). Moreover, we fuse this sparse approximation with the Nyström method, resulting in the locally corrected Nyström method (LCN). These approximations enable general log-linear time algorithms for entropy-regularized OT that perform well even in complex, high-dimensional spaces. We thoroughly demonstrate these advantages via a theoretical analysis and by evaluating multiple approximations both directly and as a component of two real-world models. Using approximate Sinkhorn for unsupervised word embedding alignment enables us to train the model full-batch in a fraction of the time while improving upon the original on average by 3.1 percentage points without any model changes. For graph distance regression we propose the graph transport network (GTN), which combines graph neural networks (GNNs) with enhanced Sinkhorn and outcompetes previous models by $48\,\%$. LCN-Sinkhorn enables GTN to achieve this while still scaling log-linearly in the number of nodes.

## 1 Introduction

Measuring the distance between two distributions or sets of objects is a central problem in machine learning. One common method of solving this is optimal transport (OT). OT is concerned with the problem of finding the transport plan for moving a source distribution (e.g. a pile of earth) to a sink distribution (e.g. a construction pit) with the cheapest cost w.r.t. some pointwise cost function (e.g. the Euclidean distance). The advantages of this method have been shown numerous times, e.g. in generative modelling (Arjovsky et al., 2017; Bousquet et al., 2017; Genevay et al., 2018), loss functions (Frogner et al., 2015), set matching (Wang et al., 2019), or domain adaptation (Courty et al., 2017). Motivated by this, many different methods for accelerating OT have been proposed in recent years (Indyk & Thaper, 2003; Papadakis et al., 2014; Backurs et al., 2020). However, most of these approaches are specialized methods that do not generalize to modern deep learning models, which rely on dynamically changing high-dimensional embeddings.

In this work we aim to make OT computation for point sets more scalable by proposing two fast and accurate approximations of entropy-regularized optimal transport: Sparse Sinkhorn and LCN-Sinkhorn, the latter relying on our newly proposed locally corrected Nyström (LCN) method. Sparse Sinkhorn uses a sparse cost matrix to leverage the fact that in entropy-regularized OT (also known as the Sinkhorn distance) (Cuturi, 2013) often only each point's nearest neighbors influence the result. LCN-Sinkhorn extends this approach by leveraging LCN, a general similarity matrix approximation that fuses local (sparse) and global (low-rank) approximations, allowing us to simultaneously capture both kinds of behavior. LCN-Sinkhorn thus fuses sparse Sinkhorn and Nyström-Sinkhorn (Altschuler et al., 2019). Both sparse Sinkhorn and LCN-Sinkhorn run in log-linear time.

We theoretically analyze these approximations and show that sparse corrections can lead to significant improvements over the Nyström approximation. We furthermore validate these approximations by showing that they are able to reproduce both the Sinkhorn distance and transport plan significantly better than previous methods across a wide range of regularization parameters and computational

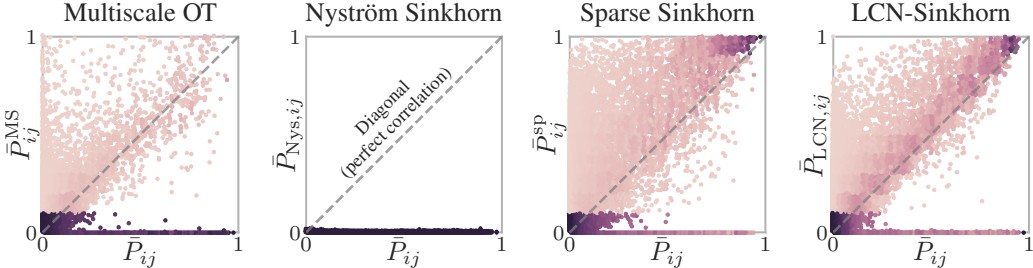

Figure 1: The proposed methods (sparse and LCN-Sinkhorn) show a clear correlation with the full Sinkhorn transport plan, as opposed to previous methods. Entries of approximations (y-axis) and full Sinkhorn (x-axis) for pre-aligned word embeddings (EN-DE). Color denotes sample density.

budgets (as e.g. demonstrated in Fig. 1). We then show the impact of these improvements by employing Sinkhorn approximations end-to-end in two high-impact machine learning tasks. First, we incorporate them into Wasserstein Procrustes for word embedding alignment (Grave et al., 2019). LCN-Sinkhorn improves upon the original method's accuracy by 3.1 percentage points using a third of the training time without *any* further model changes. Second, we develop the graph transport network (GTN), which combines graph neural networks (GNNs) with optimal transport, and further improve it via learnable unbalanced OT and multi-head OT. GTN with LCN-Sinkhorn is the first model that both overcomes the bottleneck of using a single embedding per graph and scales log-linearly in the number of nodes. In summary, our paper's main contributions are:

- Locally Corrected Nyström (LCN), a flexible, log-linear time approximation for similarity matrices, leveraging both local (sparse) and global (low-rank) approximations.
- Entropy-regularized optimal transport (a.k.a. Sinkhorn distance) with log-linear runtime via sparse Sinkhorn and LCN-Sinkhorn. These are the first log-linear approximations that are stable enough to substitute full entropy-regularized OT in models that leverage high-dimensional spaces.
- The graph transport network (GTN), which combines a graph neural network (GNN) with multi-head unbalanced LCN-Sinkhorn. GTN both sets the state of the art on graph distance regression and still scales log-linearly in the number of nodes.

## 2 SPARSE SINKHORN

**Entropy-regularized optimal transport.** In this work we focus on optimal transport between two discrete sets of points. We furthermore add entropy regularization, which enables fast computation and often performs better than regular OT (Cuturi, 2013). Formally, given two categorical distributions modelled via the vectors $\boldsymbol{p} \in \mathbb{R}^n$ and $\boldsymbol{q} \in \mathbb{R}^m$ supported on two sets of points $X_{\mathrm{p}} = \{\boldsymbol{x}_{\mathrm{p}1}, \ldots, \boldsymbol{x}_{\mathrm{p}n}\}$ and $X_{\mathrm{q}} = \{\boldsymbol{x}_{\mathrm{q}1}, \ldots, \boldsymbol{x}_{\mathrm{q}m}\}$ in $\mathbb{R}^d$ and the cost function $c : \mathbb{R}^d \times \mathbb{R}^d \to \mathbb{R}$ (e.g. the squared $L_2$ distance) giving rise to the cost matrix $\boldsymbol{C}_{ij} = c(\boldsymbol{x}_{\mathrm{p}i}, \boldsymbol{x}_{\mathrm{q}i})$ we aim to find the Sinkhorn distance $d_c^\lambda$ and the associated optimal transport plan $\bar{\boldsymbol{P}}$ (Cuturi, 2013)

$$d_c^\lambda = \min_{\boldsymbol{P}} \langle \boldsymbol{P}, \boldsymbol{C} \rangle_{\mathrm{F}} - \lambda H(\boldsymbol{P}), \qquad \text{s.t. } \boldsymbol{P}\mathbf{1}_m = \boldsymbol{p}, \ \boldsymbol{P}^T\mathbf{1}_n = \boldsymbol{q}, \tag{1}$$

with the Frobenius inner product $\langle ., . \rangle_{\mathrm{F}}$ and the entropy $H(\boldsymbol{P}) = -\sum_{i=1}^{n}\sum_{j=1}^{m} \boldsymbol{P}_{ij} \log \boldsymbol{P}_{ij}$. Note that $d_c^\lambda$ includes the entropy and can thus be negative, while Cuturi (2013) originally used $d_{\mathrm{Cuturi},c}^{1/\lambda} = \langle \bar{\boldsymbol{P}}, \boldsymbol{C} \rangle_{\mathrm{F}}$. This optimization problem can be solved by finding the vectors $\bar{\boldsymbol{s}}$ and $\bar{\boldsymbol{t}}$ that normalize the columns and rows of the matrix $\bar{\boldsymbol{P}} = \mathrm{diag}(\bar{\boldsymbol{s}})\boldsymbol{K}\,\mathrm{diag}(\bar{\boldsymbol{t}})$ with the similarity matrix $\boldsymbol{K}_{ij} = e^{-\frac{\boldsymbol{C}_{ij}}{\lambda}}$, so that $\bar{\boldsymbol{P}}\mathbf{1}_m = \boldsymbol{p}$ and $\bar{\boldsymbol{P}}^T\mathbf{1}_n = \boldsymbol{q}$. This is usually achieved via the Sinkhorn algorithm, which initializes the normalization vectors as $\boldsymbol{s}^{(1)} = \mathbf{1}_n$ and $\boldsymbol{t}^{(1)} = \mathbf{1}_m$ and then updates them alternatingly via

$$\boldsymbol{s}^{(i)} = \boldsymbol{p} \oslash (\boldsymbol{K}\boldsymbol{t}^{(i-1)}), \qquad \boldsymbol{t}^{(i)} = \boldsymbol{q} \oslash (\boldsymbol{K}^T \boldsymbol{s}^{(i)}) \tag{2}$$

until convergence, where $\oslash$ denotes elementwise division.

**Sparse Sinkhorn.** The Sinkhorn algorithm is faster than non-regularized EMD algorithms, which run in $\mathcal{O}(n^2 m \log n \log(n \max(\boldsymbol{C})))$ (Tarjan, 1997). However, its computational cost is still quadratic

in time, i.e. $\mathcal{O}(nm)$, which is prohibitively expensive for large $n$ and $m$. We propose to overcome this by observing that the matrix $\boldsymbol{K}$, and hence also $\bar{\boldsymbol{P}}$, is negligibly small everywhere except at each point's closest neighbors because of the exponential used in $\boldsymbol{K}$'s computation. We propose to leverage this by approximating $\boldsymbol{C}$ via the sparse matrix $\boldsymbol{C}^{\mathrm{sp}}$, where

$$\boldsymbol{C}_{ij}^{\mathrm{sp}} = \begin{cases} \boldsymbol{C}_{ij} & \text{if } \boldsymbol{x}_{\mathrm{p}i} \text{ and } \boldsymbol{x}_{\mathrm{q}j} \text{ are "near"}, \\ \infty & \text{otherwise.} \end{cases} \tag{3}$$

$\boldsymbol{K}^{\mathrm{sp}}$ and $\bar{\boldsymbol{P}}^{\mathrm{sp}}$ follow according to the definitions of $\boldsymbol{K}$ and $\bar{\boldsymbol{P}}$. In this work we primarily consider neighbors with distance lower than $r_1$ as "near". Finding such neighbors can be efficiently solved via locality sensitive hashing (LSH) on $X_{\mathrm{p}} \cup X_{\mathrm{q}}$.

**Locality sensitive hashing.** LSH tries to filter "near" from "far" data points by putting them into different hash buckets. Points closer than a certain distance $r_1$ are put into the same bucket with probability at least $p_1$, while those beyond some distance $r_2 = c \cdot r_1$ with $c > 1$ are put into the same bucket with probability at most $p_2 \ll p_1$. There is a plethora of LSH methods for different cost functions (Wang et al., 2014; Shrivastava & Li, 2014), so we do not have to restrict our approach to a limited set of functions. In this work we focus on cross-polytope LSH (Andoni et al., 2015) and k-means LSH (Paulevé et al., 2010), depending on the cost function (see App. H). Sparse Sinkhorn with LSH scales log-linearly with the number of points, i.e. $\mathcal{O}(n \log n)$ for $n \approx m$ (see App. A and App. K for details). Unfortunately, LSH can fail when e.g. the cost between pairs is very similar (see App. B). However, we can alleviate these limitations by fusing $\boldsymbol{K}^{\mathrm{sp}}$ with the Nyström approximation.

## 3 Locally Corrected Nyström and LCN-Sinkhorn

**Nyström method.** The Nyström method is a popular way of approximating similarity matrices that provides performance guarantees for many important tasks (Williams & Seeger, 2001; Musco & Musco, 2017). It approximates a positive semi-definite (PSD) similarity matrix $\boldsymbol{K}$ via its low-rank decomposition $\boldsymbol{K}_{\mathrm{Nys}} = \boldsymbol{U}\boldsymbol{A}^{-1}\boldsymbol{V}$. Since the optimal decomposition via SVD is too expensive to compute, Nyström instead chooses a set of $l$ landmarks $L = \{\boldsymbol{x}_{l1}, \ldots, \boldsymbol{x}_{ll}\}$ and obtains the matrices via $\boldsymbol{U}_{ij} = k(\boldsymbol{x}_{\mathrm{p}i}, \boldsymbol{x}_{lj})$, $\boldsymbol{A}_{ij} = k(\boldsymbol{x}_{li}, \boldsymbol{x}_{lj})$, and $\boldsymbol{V}_{ij} = k(\boldsymbol{x}_{li}, \boldsymbol{x}_{\mathrm{q}j})$, where $k(\boldsymbol{x}_1, \boldsymbol{x}_2)$ is an arbitrary PSD kernel, e.g. $k(\boldsymbol{x}_1, \boldsymbol{x}_2) = e^{-\frac{c(\boldsymbol{x}_1, \boldsymbol{x}_2)}{\lambda}}$ for Sinkhorn. Common methods of choosing landmarks from $X_{\mathrm{p}} \cup X_{\mathrm{q}}$ are uniform and ridge leverage score (RLS) sampling. We instead focus on k-means Nyström and sampling via k-means++, which we found to be significantly faster than recursive RLS sampling (Zhang et al., 2008) and perform better than both uniform and RLS sampling (see App. H).

**Sparse vs. Nyström.** Exponential kernels like the one used for $\boldsymbol{K}$ (e.g. the Gaussian kernel) typically have a reproducing kernel Hilbert space that is infinitely dimensional. The resulting Gram matrix $\boldsymbol{K}$ thus always has full rank. A low-rank approximation like the Nyström method can therefore only account for its global structure and not the local structure around each point $\boldsymbol{x}$. As such, it is ill-suited for any moderately low entropy regularization parameter, where the transport matrix $\bar{\boldsymbol{P}}$ resembles a permutation matrix. Sparse Sinkhorn, on the other hand, cannot account for global structure and instead approximates all non-selected distances as infinity. It will hence fail if more than a handful of neighbors are required per point. These approximations are thus opposites of each other, and as such not competing but rather *complementary* approaches.

**Locally corrected Nyström.** Since we know that the entries in our sparse approximation are exact, fusing this matrix with the Nyström method is rather straightforward. For all non-zero values in the sparse approximation $\boldsymbol{K}^{\mathrm{sp}}$ we first calculate the corresponding Nyström approximations, obtaining the sparse matrix $\boldsymbol{K}_{\mathrm{Nys}}^{\mathrm{sp}}$. To obtain the locally corrected Nyström (LCN) approximation we remove these entries from $\boldsymbol{K}_{\mathrm{Nys}}$ and replace them with their exact values, i.e.

$$\boldsymbol{K}_{\mathrm{LCN}} = \boldsymbol{K}_{\mathrm{Nys}} + \boldsymbol{K}_{\Delta}^{\mathrm{sp}} = \boldsymbol{K}_{\mathrm{Nys}} - \boldsymbol{K}_{\mathrm{Nys}}^{\mathrm{sp}} + \boldsymbol{K}^{\mathrm{sp}}. \tag{4}$$

**LCN-Sinkhorn.** To obtain the approximate transport plan $\bar{\boldsymbol{P}}_{\mathrm{LCN}}$ we run the Sinkhorn algorithm with $\boldsymbol{K}_{\mathrm{LCN}}$ instead of $\boldsymbol{K}$. However, we never fully instantiate $\boldsymbol{K}_{\mathrm{LCN}}$. Instead, we only save the decomposition and directly use these parts in Eq. (2) via $\boldsymbol{K}_{\mathrm{LCN}}\boldsymbol{t} = \boldsymbol{U}(\boldsymbol{A}^{-1}\boldsymbol{V}\boldsymbol{t}) + \boldsymbol{K}_{\Delta}^{\mathrm{sp}}\boldsymbol{t}$, similarly to Altschuler et al. (2019). As a result we obtain the decomposition of $\bar{\boldsymbol{P}}_{\mathrm{LCN}} = \bar{\boldsymbol{P}}_{\mathrm{Nys}} + \bar{\boldsymbol{P}}_{\Delta}^{\mathrm{sp}} = \bar{\boldsymbol{P}}_U \bar{\boldsymbol{P}}_W + \bar{\boldsymbol{P}}^{\mathrm{sp}} - \bar{\boldsymbol{P}}_{\mathrm{Nys}}^{\mathrm{sp}}$ and the approximate distance (using Lemma A from Altschuler et al. (2019))

$$d_{\mathrm{LCN},c}^{\lambda} = \lambda \left( \boldsymbol{s}^T \bar{\boldsymbol{P}}_U \bar{\boldsymbol{P}}_W \mathbf{1}_m + \mathbf{1}_n^T \bar{\boldsymbol{P}}_U \bar{\boldsymbol{P}}_W \boldsymbol{t} + \boldsymbol{s}^T \bar{\boldsymbol{P}}_{\Delta}^{\mathrm{sp}} \mathbf{1}_m + \mathbf{1}_n^T \bar{\boldsymbol{P}}_{\Delta}^{\mathrm{sp}} \boldsymbol{t} \right). \tag{5}$$

This approximation scales log-linearly with dataset size (see App. A and App. K for details). It allows us to smoothly move from Nyström-Sinkhorn to sparse Sinkhorn by varying the number of neighbors and landmarks. We can thus freely choose the optimal "operating point" based on the underlying problem and regularization parameter. We discuss the limitations of LCN-Sinkhorn in App. B.

## 4 THEORETICAL ANALYSIS

**Approximation error.** The main question we aim to answer in our theoretical analysis is what improvements to expect from adding sparse corrections to Nyström Sinkhorn. To do so, we first analyse approximations of $K$ in a uniform and a clustered data model. In these we use Nyström and LSH schemes that largely resemble $k$-means, as used in most of our experiments. Relevant proofs and notes for this section can be found in App. C to G.

**Theorem 1.** *Let $X_p$ and $X_q$ have $n$ samples that are uniformly distributed in a $d$-dimensional closed, locally Euclidean manifold with unit volume. Let furthermore $C_{ij} = \|x_{pi} - x_{qj}\|_2$ and $K_{ij} = e^{-C_{ij}/\lambda}$. Let the $l$ landmarks $L$ be arranged optimally and regularly so that the expected $L_2$ distance to the closest landmark is minimized. Denote $R = \frac{1}{2} \min_{x,y \in L, x \neq y} \|x - y\|_2$. Assume that the sparse correction $K_{ij}^{sp} = K_{ij}$ if and only if $x_{qj}$ is one of the $k-1$ nearest neighbors of $x_{pi}$, and that the distance to $x_{pi}$'s $k$-nearest neighbor $\delta_k \ll R$. Then the expected maximum error in row $i$ of the LCN approximation $K_{LCN}$ is*

$$\mathbb{E}[\|K_{i,:} - K_{LCN,i,:}\|_\infty] = \mathbb{E}[e^{-\delta_k/\lambda}] - \mathbb{E}[K_{LCN,i,j}], \tag{6}$$

*with $j$ denoting the index of $x_{pi}$'s $k$-nearest neighbor. Using the upper incomplete Gamma function $\Gamma(.,.)$ we can furthermore bound the second term by*

$$e^{-\sqrt{d}R/\lambda} \leq \mathbb{E}[K_{LCN,i,j}] \leq \frac{2d(\Gamma(d) - \Gamma(d, 2R/\lambda))}{(2R/\lambda)^d(1 + e^{-2R/\lambda})} + \mathcal{O}(e^{-2\sqrt{3}R/\lambda}). \tag{7}$$

The error in Eq. (6) is dominated by the first term since $\delta_k \ll R$. Note that $R$ only decreases slowly with the number of landmarks since $R \geq (\frac{(d/2)!}{l})^{1/d} \frac{1}{2\sqrt{\pi}}$ (Cohn, 2017). Moving from pure Nyström to LCN by correcting the nearest neighbors' entries thus provides significant benefits, even for uniform data. For example, by just correcting the first neighbor we obtain a $68\%$ improvement in the first term ($d = 32$, $\lambda = 0.05$, $n = 1000$). This is even more pronounced in clustered data.

**Theorem 2.** *Let $X_p, X_q \subseteq \mathbb{R}^d$ be distributed inside the same $c$ clusters with cluster centers $x_c$. Let $r$ be the maximum $L_2$ distance of a point to its cluster center and $D$ the minimum distance between two points from different clusters, with $r \ll D$. Let each LSH bucket used for the sparse approximation $K^{sp}$ cover at least one cluster. Let $K_{Nys}$ use $1 \leq l \leq d$ and $K_{LCN}$ use $l = 1$ optimally distributed landmarks per cluster. Then the maximum error is*

$$\max \|K - K_{Nys}\|_\infty = 1 - \max_{\Delta \in [0,r]} \frac{le^{-2\sqrt{r^2 + \frac{l-1}{2l}\Delta^2}/\lambda}}{1 + (l-1)e^{-\Delta/\lambda}} - \mathcal{O}(e^{-D/\lambda}), \tag{8}$$

$$\max \|K - K^{sp}\|_\infty = e^{-D/\lambda}, \tag{9}$$

$$\max \|K - K_{LCN}\|_\infty = e^{-D/\lambda}(1 - e^{-2r/\lambda}(2 - e^{-2r/\lambda}) + \mathcal{O}(e^{-D/\lambda})). \tag{10}$$

Since we can lower bound Eq. (8) by $1 - le^{-2r/\lambda} - \mathcal{O}(e^{-D/\lambda})$ we can conclude that the error in $K_{Nys}$ is close to 1 for any reasonably large $\frac{r}{\lambda}$ (which is the maximum error possible). The errors in $K^{sp}$ and $K_{LCN}$ on the other hand are vanishingly small, since $r \ll D$.

Moreover, these maximum approximation error improvements directly translate to improvements in the Sinkhorn approximation. We can show this by slightly adapting the error bounds for an approximate Sinkhorn transport plan and distance due to Altschuler et al. (2019).

**Theorem 3** (Altschuler et al. (2019)). *Let $X_p, X_q \subseteq \mathbb{R}^d$ have $n$ samples. Denote $\rho$ as the maximum distance between two samples. Let $\tilde{K}$ be an approximation of the similarity matrix $K$ with $K_{ij} = e^{-\|x_{pi} - x_{qj}\|_2/\lambda}$ and $\|\tilde{K} - K\|_\infty \leq \frac{\varepsilon'}{2} e^{-\rho/\lambda}$, where $\varepsilon' = \min(1, \frac{\varepsilon}{50(\rho + \lambda \log \frac{\lambda n}{\varepsilon})})$. When performing the Sinkhorn algorithm until $\|\tilde{P}\mathbf{1}_N - p\|_1 + \|\tilde{P}^T\mathbf{1}_N - q\|_1 \leq \varepsilon'/2$, the resulting approximate transport plan $\tilde{P}$ and distance $\tilde{d}_c^\lambda$ are bounded by*

$$|d_c^\lambda - \tilde{d}_c^\lambda| \leq \varepsilon, \qquad D_{KL}(\bar{P}\|\tilde{P}) \leq \varepsilon/\lambda. \tag{11}$$

**Convergence rate.** We next show that approximate Sinkhorn converges as fast as regular Sinkhorn by slightly adapting the convergence bound by Dvurechensky et al. (2018) to account for sparsity.

**Theorem 4** (Dvurechensky et al. (2018)). *Given the matrix* $\tilde{K} \in \mathbb{R}^{n \times n}$ *and* $p$, $q$ *the Sinkhorn algorithm gives a transport plan satisfying* $\|\tilde{P}\mathbf{1}_N - p\|_1 + \|\tilde{P}^T\mathbf{1}_N - q\|_1 \le \varepsilon$ *in iterations*

$$k \le 2 + \frac{-4\ln(\min_{i,j}\{\tilde{K}_{ij}|\tilde{K}_{ij} > 0\}\min_{i,j}\{p_i, q_j\})}{\varepsilon}. \tag{12}$$

**Backpropagation.** Efficient gradient computation is almost as important for modern deep learning models as the algorithm itself. These models usually aim at learning the embeddings in $X_\mathrm{p}$ and $X_\mathrm{q}$ and therefore need gradients w.r.t. the cost matrix $C$. We can estimate these either via automatic differentiation of the unrolled Sinkhorn iterations or via the analytic solution that assumes exact convergence. Depending on the problem at hand, either the automatic or the analytic estimator will lead to faster overall convergence (Ablin et al., 2020). LCN-Sinkhorn works flawlessly with automatic backpropagation since it only relies on basic linear algebra (except for choosing Nyström landmarks and LSH neighbors, for which we use a simple straight-through estimator (Bengio et al., 2013)). To enable fast analytic backpropagation we provide analytic gradients in Proposition 1. Note that both backpropagation methods have runtime linear in the number of points $n$ and $m$.

**Proposition 1.** *The derivatives of the distances* $d_c^\lambda$ *and* $d_{\mathrm{LCN},c}^\lambda$ *(Eqs.* (1) *and* (5)*) and the optimal transport plan* $\bar{P} \in \mathbb{R}^{n \times m}$ *w.r.t. the (decomposed) cost matrix* $C \in \mathbb{R}^{n \times m}$ *in entropy-regularized OT and LCN-Sinkhorn are*

$$\frac{\partial d_c^\lambda}{\partial C} = \bar{P}, \qquad \frac{\partial \bar{P}_{ij}}{\partial C_{kl}} = -\frac{1}{\lambda}\bar{P}_{ij}\delta_{ik}\delta_{jl}, \tag{13}$$

$$\frac{\partial d_{\mathrm{LCN},c}^\lambda}{\partial U} = -\lambda\bar{s}(W\bar{t})^T, \quad \frac{\partial d_{\mathrm{LCN},c}^\lambda}{\partial W} = -\lambda(\bar{s}^T U)^T\bar{t}^T, \quad \frac{\partial d_{\mathrm{LCN},c}^\lambda}{\partial \log K^{\mathrm{sp}}} = -\lambda\bar{P}^{\mathrm{sp}}, \quad \frac{\partial d_{\mathrm{LCN},c}^\lambda}{\partial \log K_{\mathrm{Nys}}^{\mathrm{sp}}} = -\lambda\bar{P}_{\mathrm{Nys}}^{\mathrm{sp}}, \tag{14}$$

$$\frac{\partial \bar{P}_{U,ij}}{\partial U_{kl}} = \delta_{ik}\delta_{jl}s_i, \qquad \frac{\partial \bar{P}_{W,ij}}{\partial U_{kl}} = \bar{P}_{U,ik}^\dagger s_k \bar{P}_{W,lj}, \qquad \frac{\partial \bar{P}_{U,ij}}{\partial W_{kl}} = \bar{P}_{U,ik}t_l\bar{P}_{W,lj}^\dagger,$$

$$\frac{\partial \bar{P}_{W,ij}}{\partial W_{kl}} = \delta_{ik}\delta_{jl}t_j, \qquad \frac{\partial \bar{P}_{ij}^{\mathrm{sp}}}{\partial \log K_{kl}^{\mathrm{sp}}} = \bar{P}_{ij}^{\mathrm{sp}}\delta_{ik}\delta_{jl}, \qquad \frac{\partial \bar{P}_{\mathrm{Nys},ij}^{\mathrm{sp}}}{\partial \log K_{\mathrm{Nys},kl}^{\mathrm{sp}}} = \bar{P}_{\mathrm{Nys},ij}^{\mathrm{sp}}\delta_{ik}\delta_{jl}, \tag{15}$$

*with* $\delta_{ij}$ *denoting the Kronecker delta and* $\dagger$ *the Moore-Penrose pseudoinverse. Using these decompositions we can backpropagate through LCN-Sinkhorn in time* $\mathcal{O}((n+m)l^2 + l^3)$.

## 5 GRAPH TRANSPORT NETWORK

**Graph distance learning.** The ability to predict similarities or distances between graph-structured objects is useful across a wide range of applications. It can e.g. be used to predict the reaction rate between molecules (Houston et al., 2019), search for similar images (Johnson et al., 2015), similar molecules for drug discovery (Birchall et al., 2006), or similar code for vulnerability detection (Li et al., 2019). We propose the graph transport network (GTN) to evaluate approximate Sinkhorn and advance the state of the art on this task.

**Graph transport network.** GTN first uses a Siamese graph neural network (GNN) to embed two graphs independently as *sets* of node embeddings. These embeddings are then matched using enhanced entropy-regularized optimal transport. Given an undirected graph $\mathcal{G} = (\mathcal{V}, \mathcal{E})$, with node set $\mathcal{V}$ and edge set $\mathcal{E}$, node attributes $x_i \in \mathbb{R}^{H_\mathrm{x}}$ and (optional) edge attributes $e_{i,j} \in \mathbb{R}^{H_\mathrm{e}}$, with $i,j \in \mathcal{V}$, we update the node embeddings in each GNN layer via

$$h_{\mathrm{self},i}^{(l)} = \sigma(W_{\mathrm{node}}^{(l)}h_i^{(l-1)} + b^{(l)}), \tag{16}$$

$$h_i^{(l)} = h_{\mathrm{self},i}^{(l)} + \sum_{j \in \mathcal{N}_i} \eta_{i,j}^{(l)}h_{\mathrm{self},j}^{(l)}\mathsf{W}_{\mathrm{edge}}e_{i,j}, \tag{17}$$

with $\mathcal{N}_i$ denoting the neighborhood of node $i$, $h_i^{(0)} = x_i$, $h_i^{(l)} \in \mathbb{R}^{H_\mathrm{N}}$ for $l \ge 1$, the bilinear layer $\mathsf{W}_{\mathrm{edge}} \in \mathbb{R}^{H_\mathrm{N} \times H_\mathrm{N} \times H_\mathrm{e}}$, and the degree normalization $\eta_{i,j}^{(1)} = 1$ and $\eta_{i,j}^{(l)} = 1/\sqrt{\deg_i \deg_j}$ for $l > 1$.

This choice of $\eta_{i,j}$ allows our model to handle highly skewed degree distributions while still being able to represent node degrees. We found the choice of non-linearity $\sigma$ not to be critical and chose a LeakyReLU. We do not use the bilinear layer $\mathbf{W}_{\text{edge}}e_{i,j}$ if there are no edge attributes. We aggregate each layer's node embeddings to obtain the final embedding of node $i$

$$\boldsymbol{h}_i^{\text{final}} = [\boldsymbol{h}_{\text{self},i}^{(1)} \,\|\, \boldsymbol{h}_i^{(1)} \,\|\, \boldsymbol{h}_i^{(2)} \,\|\, \dots \,\|\, \boldsymbol{h}_i^{(L)}]. \tag{18}$$

Having obtained the embedding sets $H_1^{\text{final}}$ and $H_2^{\text{final}}$ of both graphs we use the $L_2$ distance as a cost function and then calculate the Sinkhorn distance, which is symmetric and permutation invariant w.r.t. the sets $H_1^{\text{final}}$ and $H_2^{\text{final}}$. We obtain the embeddings for matching via $\boldsymbol{h}_i^{(0)} = \text{MLP}(\boldsymbol{h}_i^{\text{final}})$ and obtain the final prediction via $d = d_c^\lambda w_{\text{out}} + b_{\text{out}}$, with learnable $w_{\text{out}}$ and $b_{\text{out}}$. All weights in GTN are trained end-to-end via backpropagation. For small graphs we use the full Sinkhorn distance and scale to large graphs by leveraging LCN-Sinkhorn. GTN is more expressive than models that aggregate node embeddings to a single fixed-size embedding for the entire graph but still scales log-linearly in the number of nodes, as opposed to previous approaches that scale quadratically. Note that GTN inherently performs graph matching and can therefore also be applied to this task.

**Learnable unbalanced OT.** Since GTN regularly encounters graphs with disagreeing numbers of nodes it needs to be able to handle cases where $\|\boldsymbol{p}\|_1 \neq \|\boldsymbol{q}\|_1$ or where not all nodes in one graph have a corresponding node in the other and thus $\boldsymbol{P}\mathbf{1}_m < \boldsymbol{p}$ or $\boldsymbol{P}^T\mathbf{1}_n < \boldsymbol{q}$. Unbalanced OT allows us to handle both of these cases (Peyré & Cuturi, 2019). Previous methods did so by swapping these requirements with a uniform divergence loss term on $\boldsymbol{p}$ and $\boldsymbol{q}$ (Frogner et al., 2015; Chizat et al., 2018). However, these approaches *uniformly* penalize deviations from balanced OT and therefore cannot adapt to only ignore parts of the distribution. We propose to alleviate this limitation by swapping the cost matrix $\boldsymbol{C}$ with the bipartite matching (BP) matrix (Riesen & Bunke, 2009)

$$\boldsymbol{C}_{\text{BP}} = \begin{bmatrix} \boldsymbol{C} & \boldsymbol{C}^{(\text{p},\varepsilon)} \\ \boldsymbol{C}^{(\varepsilon,\text{q})} & \boldsymbol{C}^{(\varepsilon,\varepsilon)} \end{bmatrix}, \quad \boldsymbol{C}_{ij}^{(\text{p},\varepsilon)} = \begin{cases} c_{i,\varepsilon} & i = j \\ \infty & i \neq j \end{cases}, \quad \boldsymbol{C}_{ij}^{(\varepsilon,\text{q})} = \begin{cases} c_{\varepsilon,j} & i = j \\ \infty & i \neq j \end{cases}, \quad \boldsymbol{C}_{ij}^{(\varepsilon,\varepsilon)} = 0, \tag{19}$$

and adaptively computing the costs $c_{i,\varepsilon}$, $c_{\varepsilon,j}$ and $c_{\varepsilon,\varepsilon}$ based on the input sets $X_{\text{p}}$ and $X_{\text{q}}$. Using the BP matrix adds minor computational overhead since we only need to save the diagonals $\boldsymbol{c}_{\text{p},\varepsilon}$ and $\boldsymbol{c}_{\varepsilon,\text{q}}$ of $\boldsymbol{C}_{\text{p},\varepsilon}$ and $\boldsymbol{C}_{\varepsilon,\text{q}}$. We can then include the additional parts of $\boldsymbol{C}_{\text{BP}}$ in the Sinkhorn algorithm (Eq. (2)) via

$$\boldsymbol{K}_{\text{BP}}\boldsymbol{t} = \begin{bmatrix} \boldsymbol{K}\hat{\boldsymbol{t}} + \boldsymbol{c}_{\text{p},\varepsilon} \odot \check{\boldsymbol{t}} \\ \boldsymbol{c}_{\varepsilon,\text{q}} \odot \hat{\boldsymbol{t}} + \mathbf{1}_n^T\check{\boldsymbol{t}} \end{bmatrix}, \qquad \boldsymbol{K}_{\text{BP}}^T\boldsymbol{s} = \begin{bmatrix} \boldsymbol{K}^T\hat{\boldsymbol{s}} + \boldsymbol{c}_{\varepsilon,\text{q}} \odot \check{\boldsymbol{s}} \\ \boldsymbol{c}_{\text{p},\varepsilon} \odot \hat{\boldsymbol{s}} + \mathbf{1}_m^T\check{\boldsymbol{s}} \end{bmatrix}, \tag{20}$$

where $\hat{\boldsymbol{t}}$ denotes the upper and $\check{\boldsymbol{t}}$ the lower part of the vector $\boldsymbol{t}$. To calculate $d_c^\lambda$ we can decompose the transport plan $\boldsymbol{P}_{\text{BP}}$ in the same way as $\boldsymbol{C}_{\text{BP}}$, with a single scalar for $\boldsymbol{P}_{\varepsilon,\varepsilon}$. For GTN we obtain the deletion cost via $c_{i,\varepsilon} = \|\boldsymbol{\alpha} \odot \boldsymbol{x}_{\text{p}i}\|_2$, with a learnable vector $\boldsymbol{\alpha} \in \mathbb{R}^d$.

**Multi-head OT.** Inspired by attention models (Vaswani et al., 2017) we further improve GTN by using multiple OT heads. Using $K$ heads means that we calculate OT in parallel for $K$ separate sets of embeddings representing the same pair of objects and obtain a set of distances $\boldsymbol{d}_c^\lambda \in \mathbb{R}^K$. We can then transform these distances to a final distance prediction using a set of linear layers $\boldsymbol{h}_i^{(\text{k})} = \boldsymbol{W}^{(k)}\boldsymbol{h}_i^{\text{final}}$ for head $k$ and obtain the final prediction via $d = \text{MLP}(\boldsymbol{d}_c^\lambda)$. Note that both learnable unbalanced OT and multi-head OT might be of independent interest.

## 6 RELATED WORK

**Log-linear optimal transport.** For an overview of optimal transport and its foundations see Peyré & Cuturi (2019). On low-dimensional grids and surfaces OT can be solved using dynamical OT (Papadakis et al., 2014; Solomon et al., 2014), convolutions (Solomon et al., 2015), or embedding/hashing schemes (Indyk & Thaper, 2003; Andoni et al., 2008). In higher dimensions we can use tree-based algorithms (Backurs et al., 2020) or hashing schemes (Charikar, 2002), which are however limited to a previously fixed set of points $X_{\text{p}}, X_{\text{q}}$, on which only the distributions $\boldsymbol{p}$ and $\boldsymbol{q}$ change. For sets that change dynamically (e.g. during training) one common method of achieving log-linear runtime is a multiscale approximation of entropy-regularized OT (Schmitzer, 2019; Gerber & Maggioni, 2017). Tenetov et al. (2018) recently proposed using a low-rank approximation of the Sinkhorn similarity

Table 1: Mean and standard deviation (w.r.t. last digits, in parentheses) of relative Sinkhorn distance error, IoU of top $0.1\%$ and correlation coefficient (PCC) of OT plan entries across 5 runs. Sparse Sinkhorn and LCN-Sinkhorn consistently achieve the best approximation in all 3 measures.

| | EN-DE | | | EN-ES | | | EN-FR | | | EN-RU | | |
|---|---|---|---|---|---|---|---|---|---|---|---|---|
| | Rel. err. $d_c^\lambda$ | PCC | IoU | Rel. err. $d_c^\lambda$ | PCC | IoU | Rel. err. $d_c^\lambda$ | PCC | IoU | Rel. err. $d_c^\lambda$ | PCC | IoU |
| Factored OT | 0.318(1) | 0.044(1) | 0.019(2) | 0.332(1) | 0.037(2) | 0.026(5) | 0.326(2) | 0.038(1) | 0.034(5) | 0.281(2) | 0.055(1) | 0.025(2) |
| Multiscale OT | 0.634(11) | 0.308(14) | 0.123(5) | 0.645(14) | 0.321(6) | 0.125(12) | 0.660(17) | 0.330(9) | 0.121(7) | 0.667(16) | 0.281(19) | 0.125(9) |
| Nyström Skh. | 1.183(5) | 0.077(1) | 0.045(5) | 1.175(18) | 0.068(1) | 0.048(6) | 1.172(13) | 0.070(3) | 0.052(4) | 1.228(18) | 0.091(2) | 0.047(6) |
| Sparse Skh. | **0.233(2)** | 0.552(4) | 0.102(1) | **0.217(1)** | 0.623(4) | 0.102(1) | **0.220(1)** | 0.608(5) | 0.104(2) | **0.272(2)** | 0.446(8) | 0.090(1) |
| LCN-Sinkhorn | 0.406(15) | **0.673(12)** | **0.197(7)** | 0.368(12) | **0.736(3)** | **0.201(3)** | 0.342(5) | **0.725(4)** | **0.209(3)** | 0.465(10) | **0.623(5)** | **0.210(4)** |

matrix obtained via a semidiscrete approximation of the Euclidean distance. Altschuler et al. (2019) improved upon this approach by using the Nyström method for the approximation. These approaches still struggle with high-dimensional real-world problems, as we will show in Sec. 7.

**Sliced Wasserstein distance.** Another approach to reduce the computational complexity of optimal transport (without entropy regularization) are sliced Wasserstein distances (Rabin et al., 2011). However, they require the $L_2$ distance as a cost function and are either unstable in convergence or prohibitively expensive for high-dimensional problems ($\mathcal{O}(nd^3)$) (Meng et al., 2019).

**Fast Sinkhorn.** Another line of work has been pursuing accelerating entropy-regularized OT without changing its computational complexity w.r.t. the number of points. Original Sinkhorn requires $\mathcal{O}(1/\varepsilon^2)$ iterations (Dvurechensky et al., 2018) and Jambulapati et al. (2019) recently proposed an algorithm that reduces them to $\mathcal{O}(1/\varepsilon)$. Alaya et al. (2019) proposed to reduce the size of the Sinkhorn problem by screening out neglectable components, which allows for approximation guarantees. Genevay et al. (2016) proposed using a stochastic optimization scheme instead of Sinkhorn iterations. Essid & Solomon (2018) and Blondel et al. (2018) proposed alternative regularizations to obtain OT problems with similar runtimes as the Sinkhorn algorithm. This work is largely orthogonal to ours.

**Embedding alignment.** For an overview of cross-lingual word embedding models see Ruder et al. (2019). Unsupervised word embedding alignment was proposed by Conneau et al. (2018), with subsequent advances by Alvarez-Melis & Jaakkola (2018); Grave et al. (2019); Joulin et al. (2018).

**Graph matching and distance learning.** Most recent approaches for graph matching and graph distance learning either rely on a single fixed-dimensional graph embedding (Bai et al., 2019; Li et al., 2019), or only use attention or some other strongly simplified variant of optimal transport (Bai et al., 2019; Riba et al., 2018; Li et al., 2019). Others break permutation invariance and are thus ill-suited for this task (Ktena et al., 2017; Bai et al., 2018). So far only approaches using a single graph embedding allow faster than quadratic scaling in the number of nodes. Compared to the Sinkhorn-based image model concurrently proposed by Wang et al. (2019) GTN uses no CNN or cross-graph attention, but an enhanced GNN and embedding aggregation scheme. OT has recently been proposed for graph kernels (Maretic et al., 2019; Vayer et al., 2019), which can (to a limited extent) be used for graph matching, but not for distance learning.

## 7 EXPERIMENTS

**Approximating Sinkhorn.** We start by directly investigating different Sinkhorn approximations. To do so we compute entropy-regularized OT on pairs of $10\,000$ word embeddings from Conneau et al. (2018), which we preprocess with Wasserstein Procrustes alignment in order to obtain both close and distant neighbors. We let every method use the same total number of 40 neighbors and landmarks (LCN uses 20 each) and set $\lambda = 0.05$ (as in Grave et al. (2019)). We measure transport plan approximation quality by (a) calculating the Pearson correlation coefficient (PCC) between all entries in the approximated plan and the true $\bar{P}$ and (b) comparing the sets of $0.1\%$ largest entries in the approximated and true $\bar{P}$ using the Jaccard similarity (intersection over union, IoU). In all figures the error bars denote standard deviation across 5 runs, which is often too small to be visible.
Table 1 shows that both sparse Sinkhorn, LCN-Sinkhorn and factored OT (Forrow et al., 2019) obtain distances that are significantly closer to the true $d_c^\lambda$ than Multiscale OT and Nyström-Sinkhorn. Furthermore, the transport plan computed by sparse Sinkhorn and LCN-Sinkhorn show both a PCC and IoU that are around twice as high as Multiscale OT, while Nyström-Sinkhorn and factored OT exhibit almost no correlation. LCN-Sinkhorn performs especially well in this regard. This is also

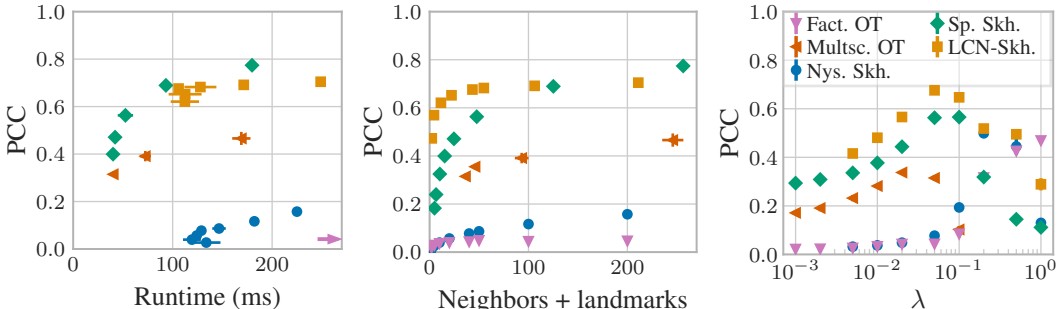

Figure 2: Tradeoff between OT plan approximation (via PCC) and runtime. Sparse Sinkhorn offers the best tradeoff, with LCN-Sinkhorn trailing closely behind. The arrow indicates factored OT results far outside the range.

Figure 3: Tradeoff between OT plan approximation and number of neighbors/landmarks. LCN-Sinkhorn achieves the best approximation for low and sparse Sinkhorn for high budgets.

Figure 4: OT plan approximation quality for varying entropy regularization $\lambda$. Sparse Sinkhorn performs best for low and LCN-Sinkhorn for moderate and factored OT for very high $\lambda$.

Table 2: Accuracy and standard deviation (w.r.t. last digits, in parentheses) across 5 runs for unsupervised word embedding alignment with Wasserstein Procrustes. LCN-Sinkhorn improves upon the original by 3.1 pp. before and 2.0 pp. after iterative CSLS refinement. *Migrated and re-run on GPU via PyTorch

| | Time (s) | EN-ES | ES-EN | EN-FR | FR-EN | EN-DE | DE-EN | EN-RU | RU-EN | Avg. |
|---|---|---|---|---|---|---|---|---|---|---|
| Original* | 268 | 79.2(2) | 78.8(2.8) | 81.0(3) | 79.4(9) | 71.7(2) | 65.7(3.4) | 36.3(1.1) | 51.1(1.1) | 67.9 |
| Full Sinkhorn | 402 | 81.1() | **82.0()** | 81.2() | 81.3() | **74.1()** | 70.7() | 37.3() | 53.5() | 70.1 |
| Multiscale OT | 88.2 | 23.6(31.4) | 74.7(3.3) | 26.9(31.7) | 6.3(4.4) | 35.8(10.4) | 47.0(20.5) | 0.0() | 0.2(1) | 26.8 |
| Nyström Skh. | 102 | 64.4(1.0) | 59.3(1.2) | 64.1(1.6) | 56.8(4.0) | 54.1(6) | 47.1(3.5) | 14.1(1.2) | 22.5(2.4) | 47.8 |
| **Sparse Skh.** | 49.2 | 80.2(2) | 81.7(4) | 80.9(3) | 80.1(2) | 72.1(6) | 65.1(1.7) | 35.5(6) | 51.5(4) | 68.4 |
| **LCN-Sinkhorn** | 86.8 | **81.8(2)** | 81.3(1.8) | 82.0(4) | **82.1(3)** | 73.6(2) | **71.3(9)** | **41.0(8)** | **55.1(1.4)** | **71.0** |
| Original* + ref. | 268+81 | 83.0(3) | 82.0(2.5) | 83.8(1) | 83.0(4) | **77.3(3)** | 69.7(4.3) | 46.2(1.0) | 54.0(1.1) | 72.4 |
| **LCN-Skh.** + ref. | 86.8+81 | **83.5(2)** | **83.1(1.3)** | 83.8(2) | **83.6(1)** | 77.2(3) | **72.8(7)** | **51.8(2.6)** | **59.2(1.9)** | **74.4** |

evident in Fig. 1, which shows how the $10^4 \times 10^4$ approximated OT plan entries compared to the true Sinkhorn values.

Fig. 2 shows that sparse Sinkhorn offers the best trade-off between runtime and OT plan quality. Factored OT exhibits a runtime 2 to 10 times longer than the competition due to its iterative refinement scheme. LCN-Sinkhorn performs best for use cases with constrained memory (few neighbors/landmarks), as shown in Fig. 3. The number of neighbors and landmarks directly determines memory usage and is linearly proportional to the runtime (see App. K). Fig. 9 shows that sparse Sinkhorn performs best for low regularizations, where LCN-Sinkhorn fails due to the Nyström part going out of bounds. Nyström Sinkhorn performs best at high values and LCN-Sinkhorn always performs better than both (as long as it can be calculated). Interestingly, all approximations except factored OT seem to fail at high $\lambda$. We defer analogously discussing the distance approximation to App. L. All approximations scale linearly both in the number of neighbors/landmarks and dataset size, as shown in App. K. Overall, we see that sparse Sinkhorn and LCN-Sinkhorn yield significant improvements over previous approximations. However, do these improvements also translate to better performance on downstream tasks?

**Embedding alignment.** Embedding alignment is the task of finding the orthogonal matrix $\boldsymbol{R} \in \mathbb{R}^{d \times d}$ that best aligns the vectors from two different embedding spaces, which is e.g. useful for unsupervised word translation. We use the experimental setup established by Conneau et al. (2018) by migrating Grave et al. (2019)'s implementation to PyTorch. The only change we make is using the full set of 20 000 word embeddings and training for 300 steps, while reducing the learning rate by half every 100 steps. We do not change *any* other hyperparameters and do not use unbalanced OT. After training we match pairs via cross-domain similarity local scaling (CSLS) (Conneau et al., 2018). We use 10 Sinkhorn iterations, 40 neighbors for sparse Sinkhorn, and 20 neighbors and landmarks for LCN-Sinkhorn (for details see App. H). We allow both multiscale OT and Nyström Sinkhorn to use as many landmarks and neighbors as can fit into GPU memory and finetune both methods.

Table 2 shows that using full Sinkhorn yields a significant improvement in accuracy on this task

Table 3: RMSE for GED regression across 3 runs and the targets' standard deviation $\sigma$. GTN outperforms previous models by 48 %.

| | Linux | AIDS30 | Pref. att. |
|---|---|---|---|
| $\sigma$ | 0.184 | 16.2 | 48.3 |
| SiamMPNN | 0.090(7) | 13.8(3) | 12.1(6) |
| SimGNN | 0.039 | 4.5(3) | 8.3(1.4) |
| GMN | 0.015() | 10.3(6) | 7.8(3) |
| GTN, 1 head | 0.022(1) | 3.7(1) | 4.5(3) |
| 8 OT heads | **0.012(1)** | **3.2(1)** | **3.6(2)** |
| Balanced OT | 0.034(1) | 15.3(1) | 27.4(9) |

Table 4: RMSE for graph distance regression across 3 runs. Using LCN-Sinkhorn with GTN increases the error by only 10 % and allows log-linear scaling.

| | GED | | PM [$10^{-2}$] |
|---|---|---|---|
| | AIDS30 | Pref. att. | Pref. att. 200 |
| $\sigma$ | 16.2 | 48.3 | 10.2 |
| Full Sinkhorn | 3.7(1) | 4.5(3) | 1.27(6) |
| Nyström Skh. | 3.6(3) | 6.2(6) | 2.43(7) |
| Multiscale OT | 11.2(3) | 27.4(5.4) | 6.71(44) |
| Sparse Skh. | 44.0(30.4) | 40.7(8.1) | 7.57(1.09) |
| LCN-Skh. | 4.0(1) | 5.1(4) | 1.41(15) |

compared to the original approach of performing Sinkhorn on randomly sampled subsets of embeddings (Grave et al., 2019). LCN-Sinkhorn even outperforms the *full* version in most cases, which is likely due to regularization effects from the approximation. It also runs 4.6x faster than full Sinkhorn and 3.1x faster than the original scheme. Sparse Sinkhorn runs 1.8x faster than LCN-Sinkhorn but cannot match its accuracy. LCN-Sinkhorn still outcompetes the original method after refining the embeddings with iterative local CSLS (Conneau et al., 2018). Both multiscale OT and Nyström Sinkhorn fail at this task, despite their larger computational budget. This shows that the improvements achieved by sparse Sinkhorn and LCN-Sinkhorn have an even larger impact in practice.

**Graph distance regression.** The graph edit distance (GED) is useful for various tasks, such as image retrieval (Xiao et al., 2008) or fingerprint matching (Neuhaus & Bunke, 2004), but its computation is NP-complete (Bunke & Shearer, 1998). Therefore, to use it on larger graphs we need to learn an approximation. We use the Linux dataset by Bai et al. (2019) and generate 2 new datasets by computing the exact GED using the method by Lerouge et al. (2017) on small graphs ($\leq 30$ nodes) from the AIDS dataset (Riesen & Bunke, 2008) and a set of preferential attachment graphs. We compare GTN to 3 state-of-the-art baselines: SiameseMPNN (Riba et al., 2018), SimGNN (Bai et al., 2019), and the Graph Matching Network (GMN) (Li et al., 2019). We tune the hyperparameters of all baselines and GTN on the validation set via a grid search. For more details see App. H to J.

We first test both GTN and the proposed OT enhancements. Table 3 shows that GTN improves upon competing models by 20 % with a single head and by 48 % with 8 OT heads. These improvements break down when using regular balanced OT, showing the importance of learnable unbalanced OT. Having established GTN as a state-of-the-art model we next ask whether we can sustain its performance when using approximate OT. To test this we additionally generate a set of larger graphs with around 200 nodes and use the Pyramid matching (PM) kernel (Nikolentzos et al., 2017) as the prediction target, since these graphs are too large to compute the GED. See App. J for hyperparameter details. Table 4 shows that both sparse Sinkhorn and the multiscale method using 4 (expected) neighbors fail at this task, demonstrating that the low-rank approximation in LCN has a crucial stabilizing effect during training. Nyström Sinkhorn with 4 landmarks performs surprisingly well on the AIDS30 dataset, suggesting an overall low-rank structure with Nyström acting as regularization. However, it does not perform as well on the other two datasets. Using LCN-Sinkhorn with 2 neighbors and landmarks works well on all three datasets, with an RMSE increased by only 10 % compared to full GTN. App. K furthermore shows that GTN with LCN-Sinkhorn indeed scales linearly in the number of nodes across multiple orders of magnitude. This model thus allows to perform graph matching and distance learning on graphs that are considered large even for simple node-level tasks (20 000 nodes).

## 8 CONCLUSION

Locality sensitive hashing (LSH) and the novel locally corrected Nyström (LCN) method enable fast and accurate approximations of entropy-regularized OT with log-linear runtime: Sparse Sinkhorn and LCN-Sinkhorn. The graph transport network (GTN) is one example for such a model, which can be substantially improved with learnable unbalanced OT and multi-head OT. It sets the new state of the art for graph distance learning while still scaling log-linearly with graph size. These contributions enable new applications and models that are both faster and more accurate, since they can sidestep workarounds such as pooling.

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

## A COMPLEXITY ANALYSIS

**Sparse Sinkhorn.** A common way of achieving a high $p_1$ and low $p_2$ in LSH is via the AND-OR construction. In this scheme we calculate $B \cdot r$ hash functions, divided into $B$ sets (hash bands) of $r$ hash functions each. A pair of points is considered as neighbors if any hash band matches completely. Calculating the hash buckets for all points with $b$ hash buckets per function scales as $\mathcal{O}((n+m)dBbr)$ for the hash functions we consider. As expected, for the tasks and hash functions we investigated we obtain approximately $m/b^r$ and $n/b^r$ neighbors, with $b^r$ hash buckets per band. Using this we can fix the number of neighbors to a small, constant $\beta$ in expectation with $b^r = \min(n,m)/\beta$. We thus obtain a sparse cost matrix $\boldsymbol{C}^{\mathrm{sp}}$ with $\mathcal{O}(\max(n,m)\beta)$ non-infinite values and can calculate $\boldsymbol{s}$ and $\boldsymbol{t}$ in linear time $\mathcal{O}(N_{\mathrm{sink}}\max(n,m)\beta)$, where $N_{\mathrm{sink}} \leq 2 + \frac{-4\ln(\min_{i,j}\{\tilde{\boldsymbol{K}}_{ij}|\tilde{\boldsymbol{K}}_{ij}>0\}\min_{i,j}\{\boldsymbol{p}_i,\boldsymbol{q}_j\})}{\varepsilon}$ (see Theorem 4) denotes the number of Sinkhorn iterations. Calculating the hash buckets with $r = \frac{\log\min(n,m)-\log\beta}{\log b}$ takes $\mathcal{O}((n+m)dBb(\log\min(n,m) - \log\beta)/\log b)$. Since $B$, $b$, and $\beta$ are small, we obtain roughly log-linear scaling with the number of points overall, i.e. $\mathcal{O}(n\log n)$ for $n \approx m$.

**LCN-Sinkhorn.** Both choosing landmarks via k-means++ sampling and via k-means with a fixed number of iterations have the same runtime complexity of $\mathcal{O}((n+m)ld)$. Precomputing $\boldsymbol{W}$ can be done in time $\mathcal{O}(nl^2 + l^3)$. The low-rank part of updating the vectors $\boldsymbol{s}$ and $\boldsymbol{t}$ can be computed in $\mathcal{O}(nl + l^2 + lm)$, with $l$ chosen constant, i.e. independently of $n$ and $m$. Since sparse Sinkhorn with LSH has a log-linear runtime we again obtain log-linear overall runtime for LCN-Sinkhorn.

## B LIMITATIONS

**Sparse Sinkhorn.** Using a sparse approximation for $\boldsymbol{K}$ works well in the common case when the regularization parameter $\lambda$ is low and the cost function varies enough between data pairs, such that the transport plan $\boldsymbol{P}$ resembles a sparse matrix. However, it can fail if the cost between pairs is very similar or the regularization is very high, if the dataset contains many hubs, i.e. points with a large number of neighbors, or if the distributions $\boldsymbol{p}$ or $\boldsymbol{q}$ are spread very unevenly. Furthermore, sparse Sinkhorn can be too unstable to train a model from scratch, since randomly initialized embeddings often have no close neighbors (see Sec. 7). LCN-Sinkhorn largely alleviates these limitations.

**LCN-Sinkhorn.** Since we cannot calculate the full cost matrix, LCN-Sinkhorn cannot provide accuracy guarantees in general. Highly concentrated distributions $\boldsymbol{p}$ and $\boldsymbol{q}$ might have adverse effects on LCN-Sinkhorn. However, we can compensate for these by sampling landmarks or neighbors proportional to each point's probability mass. We therefore do not expect LCN-Sinkhorn to break down in this scenario. If the regularization parameter is low or the cost function varies greatly, we sometimes observed stability issues (over- and underflows) with the Nyström approximation because of the inverse $\boldsymbol{A}^{-1}$, which cannot be calculated in log-space. Due to its linearity the Nyström method furthermore sometimes approximates similarities as negative values, which leads to a failure if the result of the matrix product in Eq. (2) becomes negative. In these extreme cases we also observed catastrophic elimination caused by the correction $\boldsymbol{K}_{\Delta}^{\mathrm{sp}}$. Since this essentially means that optimal transport will be very local, we recommend using sparse Sinkhorn in these scenarios. This again demonstrates the complementarity of the sparse approximation and Nyström: In cases where one fails we can often resort to the other.

## C PROOF OF THEOREM 1

We first prove a lemma that will be useful later on.

**Lemma A.** *Let $\tilde{\boldsymbol{K}}$ be the Nyström approximation of the similarity matrix $\boldsymbol{K}_{ij} = e^{-\|\boldsymbol{x}_i - \boldsymbol{x}_j\|_2/\lambda}$. Let $\boldsymbol{x}_i$ and $\boldsymbol{x}_j$ be data points with equal $L_2$ distance $r_i$ and $r_j$ to all $l$ landmarks, which have the same distance $\Delta$ to each other. Then*

$$\tilde{\boldsymbol{K}}_{ij} = \frac{le^{-(r_i+r_j)/\lambda}}{1 + (l-1)e^{-\Delta/\lambda}} \tag{21}$$

*Proof.* The inter-landmark distance matrix is

$$\boldsymbol{A} = e^{-\Delta/\lambda}\mathbf{1}_{l\times l} + (1 - e^{-\Delta/\lambda})\boldsymbol{I}_l, \tag{22}$$

where $\mathbf{1}_{l \times l}$ denotes the constant 1 matrix. Using the identity

$$(b\mathbf{1}_{n \times n} + (a-b)\mathbf{I}_n)^{-1} = \frac{-b}{(a-b)(a+(n-1)b)}\mathbf{1}_{n \times n} + \frac{1}{a-b}\mathbf{I}_n \qquad (23)$$

we compute

$$\tilde{\mathbf{K}}_{ij} = \mathbf{U}_{i,:}\mathbf{A}^{-1}\mathbf{V}_{:,j}$$

$$= \begin{pmatrix} e^{-r_i/\lambda} & e^{-r_i/\lambda} & \cdots \end{pmatrix} \left( \frac{-e^{-\Delta/\lambda}}{(1-e^{-\Delta/\lambda})(1+(l-1)e^{-\Delta/\lambda})}\mathbf{1}_{l \times l} + \frac{1}{1-e^{-\Delta/\lambda}}\mathbf{I}_l \right) \begin{pmatrix} e^{-r_j/\lambda} \\ e^{-r_j/\lambda} \\ \vdots \end{pmatrix}$$

$$= \frac{e^{-(r_i+r_j)/\lambda}}{1-e^{-\Delta/\lambda}}\left( \frac{-l^2 e^{-\Delta/\lambda}}{1+(l-1)e^{-\Delta/\lambda}} + l \right) = \frac{e^{-(r_i+r_j)/\lambda}}{1-e^{-\Delta/\lambda}}\frac{l-le^{-\Delta/\lambda}}{1+(l-1)e^{-\Delta/\lambda}}$$

$$= \frac{le^{-(r_i+r_j)/\lambda}}{1+(l-1)e^{-\Delta/\lambda}}$$

$$(24)$$

$\square$

Now consider the error $\|\mathbf{K}_{i,:} - \mathbf{K}_{\text{LCN},i,:}\|_\infty$. The $k-1$ nearest neighbors are covered by the sparse correction and therefore the next nearest neighbor has distance $\delta_k$. The expected distance from the closest landmark is greater than the expected distance inside the surrounding $d$-ball of radius $R$, i.e. $\mathbb{E}[r] \geq \mathbb{E}_{V(R)}[r] = \frac{d}{d+1}R$. Because furthermore $\delta_k \ll R$, the error is dominated by the first term and the maximum error in row $i$ is given by the $k$-nearest neighbor of $i$, denoted by $j$. Thus

$$\mathbb{E}[\|\mathbf{K}_{i,:} - \mathbf{K}_{\text{LCN},i,:}\|_\infty] = \mathbb{E}[\mathbf{K}_{i,j} - \mathbf{K}_{\text{LCN},i,j}] = \mathbb{E}[\mathbf{K}_{i,j}] - \mathbb{E}[\mathbf{K}_{\text{LCN},i,j}] = \mathbb{E}[e^{-\delta_k/\lambda}] - \mathbb{E}[\mathbf{K}_{\text{LCN},i,j}]$$
$$(25)$$

Note that we can lower bound the first term using Jensen's inequality. However, we were unable to find a reasonably tight upper bound and the resulting integral (ignoring exponentially small boundary effects, see Percus & Martin (1998))

$$\mathbb{E}[e^{-\delta_k/\lambda}] = \frac{n!}{(n-k)!(k-1)!}\int_0^{\frac{((d/2)!)^{1/d}}{\sqrt{\pi}}} e^{-r/\lambda}V(r)^{k-1}(1-V(r))^{n-k}\frac{\mathrm{d}V(r)}{\mathrm{d}r}\,\mathrm{d}r, \qquad (26)$$

with the volume of the $d$-ball

$$V(r) = \frac{\pi^{d/2}r^d}{(d/2)!} \qquad (27)$$

does not have an analytical solution. We thus have to resort to calculating this expectation numerically.

We lower bound the second term by (1) ignoring every landmark except the closest one, since additional landmarks can only increase the estimate $\mathbf{K}_{\text{LCN},i,j}$. We then (2) upper bound the $L_2$ distance to the closest landmark $r$ by $\sqrt{d}R/2$, since this would be the furthest distance to the closest point in a $d$-dimensional grid. Any optimal arrangement minimizing $\mathbb{E}[\min_{\mathbf{y} \in L}\|\mathbf{x} - \mathbf{y}\|_2 \mid \mathbf{x} \in X_\mathrm{p}]$ would be at least as good as a grid and thus have furthest distances as small or smaller than those in a grid. Thus,

$$\mathbb{E}[\mathbf{K}_{\text{LCN},i,j}] \overset{(1)}{\geq} e^{-2r/\lambda} \overset{(2)}{\geq} e^{-\sqrt{d}R/\lambda}. \qquad (28)$$

We upper bound this expectation by considering that any point outside the inscribed sphere of the space closest to a landmark (which has radius $R$) would be further away from the landmarks and thus have a lower value $e^{-d/\lambda}$. We can therefore reduce the space over which the expectation is taken to the ball with radius $R$, i.e.

$$\mathbb{E}[\mathbf{K}_{\text{LCN},i,j}] \leq \mathbb{E}_{V(R)}[\mathbf{K}_{\text{LCN},i,j}] \qquad (29)$$

Next we (1) ignore the contributions of all landmarks except for the closest 2, since a third landmark must be further away from the data point than $\sqrt{3}R$, adding an error of $\mathcal{O}(e^{-2\sqrt{3}R/\lambda})$. We then (2)

lower bound the distances of both points to both landmarks by the closest distance to a landmark $r = \min\{\|\boldsymbol{x}_i - \boldsymbol{x}_{l_1}\|_2, \|\boldsymbol{x}_i - \boldsymbol{x}_{l_2}\|_2, \|\boldsymbol{x}_j - \boldsymbol{x}_{l_1}\|_2, \|\boldsymbol{x}_j - \boldsymbol{x}_{l_2}\|_2\}$ and use Lemma A to obtain

$$
\begin{aligned}
\mathbb{E}_{V(R)}[\boldsymbol{K}_{\mathrm{LCN},i,j}] &\overset{(1)}{=} \mathbb{E}_{V(R)}[\boldsymbol{K}_{\mathrm{LCN,\,2\,landmarks},i,j}] + \mathcal{O}(e^{-2\sqrt{3}R/\lambda}) \\
&\overset{(2)}{\leq} \mathbb{E}_{V(R)}\left[\frac{2e^{-2r/\lambda}}{1 + e^{-2R/\lambda}}\right] + \mathcal{O}(e^{-2\sqrt{3}R/\lambda}) \\
&= \frac{2\mathbb{E}_{V(R)}[e^{-2r/\lambda}]}{1 + e^{-2R/\lambda}} + \mathcal{O}(e^{-2\sqrt{3}R/\lambda}).
\end{aligned}
\tag{30}
$$

Assuming Euclideanness in $V(R)$ we obtain

$$
\begin{aligned}
\mathbb{E}_{V(R)}[e^{-2r/\lambda}] &= \frac{1}{V(R)}\int_0^R e^{-2r/\lambda}\frac{\mathrm{d}V(r)}{\mathrm{d}r}\,\mathrm{d}r = \frac{d}{R}\int_0^R e^{-2r/\lambda}r^{d-1}\,\mathrm{d}r \\
&= \frac{d}{(2R/\lambda)^d}\left(\Gamma(d) - \Gamma(d, 2R/\lambda)\right)
\end{aligned}
\tag{31}
$$

$\square$

# D  PROOF OF THEOREM 2

Note that this theorem does not use probabilistic arguments but rather geometrically analyzes the maximum possible error. $\boldsymbol{K}^{\mathrm{sp}}$ is correct for all pairs inside a cluster and 0 otherwise. We therefore obtain the maximum error by considering the closest possible pair between clusters. By definition, this pair has distance $D$ and thus

$$
\max\|\boldsymbol{K} - \boldsymbol{K}^{\mathrm{sp}}\|_\infty = e^{-D/\lambda}
\tag{32}
$$

LCN is also correct for all pairs inside a cluster, so we again consider the closest possible pair $\boldsymbol{x}_i$, $\boldsymbol{x}_j$ between clusters. We furthermore only consider the landmarks of the two concerned clusters, adding an error of $\mathcal{O}(e^{-D/\lambda})$. Hence,

$$
\begin{aligned}
\boldsymbol{K}_{\mathrm{LCN,\,2\,landmarks},ij} &= \begin{pmatrix} e^{-r/\lambda} & e^{-(r+D)/\lambda} \end{pmatrix} \begin{pmatrix} 1 & e^{-(2r+D)/\lambda} \\ e^{-(2r+D)/\lambda} & 1 \end{pmatrix}^{-1} \begin{pmatrix} e^{-(r+D)/\lambda} \\ e^{-r/\lambda} \end{pmatrix} \\
&= \frac{1}{1 - e^{-(4r+2D)/\lambda}}\begin{pmatrix} e^{-r/\lambda} & e^{-(r+D)/\lambda} \end{pmatrix}\begin{pmatrix} 1 & -e^{-(2r+D)/\lambda} \\ -e^{-(2r+D)/\lambda} & 1 \end{pmatrix}\begin{pmatrix} e^{-(r+D)/\lambda} \\ e^{-r/\lambda} \end{pmatrix} \\
&= \frac{1}{1 - e^{-(4r+2D)/\lambda}}\begin{pmatrix} e^{-r/\lambda} & e^{-(r+D)/\lambda} \end{pmatrix}\begin{pmatrix} e^{-(r+D)/\lambda} - e^{-(3r+D)/\lambda} \\ e^{-r/\lambda} - e^{-(3r+2D)/\lambda} \end{pmatrix} \\
&= \frac{1}{1 - e^{-(4r+2D)/\lambda}}\left(e^{-(2r+D)/\lambda} - e^{-(4r+D)/\lambda} + e^{-(2r+D)/\lambda} - e^{-(4r+3D)/\lambda}\right) \\
&= \frac{e^{-(2r+D)/\lambda}}{1 - e^{-(4r+2D)/\lambda}}\left(2 - e^{-2r/\lambda} - e^{-(2r+2D)/\lambda}\right) \\
&= e^{-D/\lambda}e^{-2r/\lambda}(2 - e^{-2r/\lambda}) - \mathcal{O}(e^{-2D/\lambda})
\end{aligned}
\tag{33}
$$

and thus

$$
\max\|\boldsymbol{K} - \boldsymbol{K}_{\mathrm{LCN}}\|_\infty = e^{-D/\lambda}(1 - e^{-2r/\lambda}(2 - e^{-2r/\lambda}) + \mathcal{O}(e^{-D/\lambda})).
\tag{34}
$$

For pure Nyström we need to consider the distances inside a cluster. In the worst case two points overlap, i.e. $\boldsymbol{K}_{ij} = 1$, and lie at the boundary of the cluster. Since $r \ll D$ we again only consider the landmarks in the concerned cluster, adding an error of $\mathcal{O}(e^{-D/\lambda})$. Because of symmetry we can optimize the worst-case distance from all landmarks by putting them on a $(l-1)$-simplex centered on the cluster center. Since there are at most $d$ landmarks in each cluster there is always one direction in which the worst-case points are $r$ away from all landmarks. The circumradius of an $(l-1)$-simplex with side length $\Delta$ is $\sqrt{\frac{l-1}{2l}}\Delta$. Thus, the maximum distance to all landmarks is $\sqrt{r^2 + \frac{l-1}{2l}\Delta^2}$. Using Lemma A we therefore obtain the Nyström approximation

$$
\boldsymbol{K}_{\mathrm{Nys},ij} = \frac{le^{-2\sqrt{r^2 + \frac{l-1}{2l}\Delta^2}/\lambda}}{1 + (l-1)e^{-\Delta/\lambda}} + \mathcal{O}(e^{-D/\lambda})
\tag{35}
$$

$\square$

## E    NOTE ON THEOREM 3

Lemmas C-F and and thus Theorem 1 by Altschuler et al. (2019) are also valid for $\boldsymbol{Q}$ outside the simplex so long as $\|\boldsymbol{Q}\|_1 = n$ and it only has non-negative entries. Any $\tilde{\boldsymbol{P}}$ returned by Sinkhorn fulfills these conditions. Therefore the rounding procedure given by their Algorithm 4 is not necessary for this result.

Furthermore, to be more consistent with Theorems 1 and 2 we use the $L_2$ distance instead of $L_2^2$ in this theorem, which only changes the dependence on $\rho$.

## F    NOTES ON THEOREM 4

To adapt Theorem 1 by Dvurechensky et al. (2018) to sparse matrices (i.e. matrices with some $\boldsymbol{K}_{ij} = 0$) we need to redefine

$$\nu := \min_{i,j}\{\boldsymbol{K}_{ij}|\boldsymbol{K}_{ij} > 0\}, \tag{36}$$

i.e. take the minimum only w.r.t. non-zero elements in their Lemma 1.

## G    PROOF OF PROPOSITION 1

**Theorem A** (Danskin's theorem). *Consider a continuous function $\phi : \mathbb{R}^k \times Z \to \mathbb{R}$, with the compact set $Z \subset \mathbb{R}^j$. If $\phi(\boldsymbol{x}, \boldsymbol{z})$ is convex in $\boldsymbol{x}$ for every $\boldsymbol{z} \in Z$ and $\phi(\boldsymbol{x}, \boldsymbol{z})$ has a unique maximizer $\bar{\boldsymbol{z}}$, the derivative of*

$$f(\boldsymbol{x}) = \max_{\boldsymbol{z} \in Z} \phi(\boldsymbol{x}, \boldsymbol{z}) \tag{37}$$

*is given by the derivative at the maximizer, i.e.*

$$\frac{\partial f}{\partial \boldsymbol{x}} = \frac{\partial \phi(\boldsymbol{x}, \bar{\boldsymbol{z}})}{\partial \boldsymbol{x}}. \tag{38}$$

We start by deriving the derivatives of the distances. To show that the Sinkhorn distance fulfills the conditions for Danskin's theorem we first identify $\boldsymbol{x} = \boldsymbol{C}$, $\boldsymbol{z} = \boldsymbol{P}$, and $\phi(\boldsymbol{C}, \boldsymbol{P}) = -\langle \boldsymbol{P}, \boldsymbol{C} \rangle_{\mathrm{F}} + \lambda H(\boldsymbol{P})$. We next observe that the restrictions $\boldsymbol{P}\mathbf{1}_m = \boldsymbol{p}$ and $\boldsymbol{P}^T\mathbf{1}_n = \boldsymbol{q}$ define a compact, convex set for $\boldsymbol{P}$. Furthermore, $\phi$ is a continuous function and linear in $\boldsymbol{C}$, i.e. both convex and concave for any finite $\boldsymbol{P}$. Finally, $\phi(\boldsymbol{C}, \boldsymbol{P})$ is concave in $\boldsymbol{P}$ since $\langle \boldsymbol{P}, \boldsymbol{C} \rangle_{\mathrm{F}}$ is linear and $\lambda H(\boldsymbol{P})$ is concave. Therefore the maximizer $\bar{\boldsymbol{P}}$ is unique and Danskin's theorem applies to the Sinkhorn distance. Using

$$\frac{\partial \boldsymbol{C}_{\mathrm{Nys},ij}}{\partial \boldsymbol{U}_{kl}} = \frac{\partial}{\partial \boldsymbol{U}_{kl}}\left(-\lambda \log(\sum_a \boldsymbol{U}_{ia}\boldsymbol{W}_{aj})\right) = -\lambda \delta_{ik}\frac{\boldsymbol{W}_{lj}}{\sum_a \boldsymbol{U}_{ia}\boldsymbol{W}_{aj}} = -\lambda \delta_{ik}\frac{\boldsymbol{W}_{lj}}{\boldsymbol{K}_{\mathrm{Nys},ij}}, \tag{39}$$

$$\frac{\partial \boldsymbol{C}_{\mathrm{Nys},ij}}{\partial \boldsymbol{W}_{kl}} = \frac{\partial}{\partial \boldsymbol{W}_{kl}}\left(-\lambda \log(\sum_a \boldsymbol{U}_{ia}\boldsymbol{W}_{aj})\right) = -\lambda \delta_{jl}\frac{\boldsymbol{U}_{ik}}{\sum_a \boldsymbol{U}_{ia}\boldsymbol{W}_{aj}} = -\lambda \delta_{jl}\frac{\boldsymbol{U}_{ik}}{\boldsymbol{K}_{\mathrm{Nys},ij}}, \tag{40}$$

$$\frac{\bar{\boldsymbol{P}}_{\mathrm{Nys},ij}}{\boldsymbol{K}_{\mathrm{Nys},ij}} = \frac{\sum_b \bar{\boldsymbol{P}}_{U,ib}\bar{\boldsymbol{P}}_{W,bj}}{\sum_a \boldsymbol{U}_{ia}\boldsymbol{W}_{aj}} = \frac{\bar{\boldsymbol{s}}_i\bar{\boldsymbol{t}}_j \sum_b \boldsymbol{U}_{ib}\boldsymbol{W}_{bj}}{\sum_a \boldsymbol{U}_{ia}\boldsymbol{W}_{aj}} = \bar{\boldsymbol{s}}_i\bar{\boldsymbol{t}}_j\frac{\sum_b \boldsymbol{U}_{ib}\boldsymbol{W}_{bj}}{\sum_a \boldsymbol{U}_{ia}\boldsymbol{W}_{aj}} = \bar{\boldsymbol{s}}_i\bar{\boldsymbol{t}}_j \tag{41}$$

and the chain rule we can calculate the derivative w.r.t. the cost matrix as

$$\frac{\partial d_c^\lambda}{\partial \boldsymbol{C}} = -\frac{\partial}{\partial \boldsymbol{C}} \left( -\langle \bar{\boldsymbol{P}}, \boldsymbol{C} \rangle_{\mathrm{F}} + \lambda H(\bar{\boldsymbol{P}}) \right) = \bar{\boldsymbol{P}}, \tag{42}$$

$$\begin{aligned}
\frac{\partial d_{\mathrm{LCN},c}^\lambda}{\partial \boldsymbol{U}_{kl}} &= \sum_{i,j} \frac{\partial \boldsymbol{C}_{\mathrm{Nys},ij}}{\partial \boldsymbol{U}_{kl}} \frac{\partial d_{\mathrm{LCN},c}^\lambda}{\partial \boldsymbol{C}_{\mathrm{Nys},ij}} = -\lambda \sum_{i,j} \delta_{ik} \boldsymbol{W}_{lj} \frac{\bar{\boldsymbol{P}}_{\mathrm{Nys},ij}}{\boldsymbol{K}_{\mathrm{Nys},ij}} \\
&= -\lambda \sum_{i,j} \delta_{ik} \boldsymbol{W}_{lj} \bar{\boldsymbol{s}}_i \bar{\boldsymbol{t}}_j = -\lambda \bar{\boldsymbol{s}}_k \sum_j \boldsymbol{W}_{lj} \bar{\boldsymbol{t}}_j = \left( -\lambda \bar{\boldsymbol{s}} (\boldsymbol{W} \bar{\boldsymbol{t}})^T \right)_{kl},
\end{aligned} \tag{43}$$

$$\begin{aligned}
\frac{\partial d_{\mathrm{LCN},c}^\lambda}{\partial \boldsymbol{W}_{kl}} &= \sum_{i,j} \frac{\partial \boldsymbol{C}_{\mathrm{Nys},ij}}{\partial \boldsymbol{W}_{kl}} \frac{\partial d_{\mathrm{LCN},c}^\lambda}{\partial \boldsymbol{C}_{\mathrm{Nys},ij}} = -\lambda \sum_{i,j} \delta_{jl} \boldsymbol{U}_{ik} \frac{\bar{\boldsymbol{P}}_{\mathrm{Nys},ij}}{\boldsymbol{K}_{\mathrm{Nys},ij}} \\
&= -\lambda \sum_{i,j} \delta_{jl} \boldsymbol{U}_{ik} \bar{\boldsymbol{s}}_i \bar{\boldsymbol{t}}_j = -\lambda \left( \sum_i \bar{\boldsymbol{s}}_i \boldsymbol{U}_{ik} \right) \bar{\boldsymbol{t}}_l = \left( -\lambda (\bar{\boldsymbol{s}}^T \boldsymbol{U})^T \bar{\boldsymbol{t}}^T \right)_{kl},
\end{aligned} \tag{44}$$

and $\frac{\partial d_{\mathrm{LCN},c}^\lambda}{\partial \log \boldsymbol{K}^{\mathrm{sp}}}$ and $\frac{\partial d_{\mathrm{LCN},c}^\lambda}{\partial \log \boldsymbol{K}_{\mathrm{Nys}}^{\mathrm{sp}}}$ directly follow from $\frac{\partial d_c^\lambda}{\partial \boldsymbol{C}}$.

**Theorem B** (Implicit function theorem). *Let $f : \mathbb{R}^{n'} \times \mathbb{R}^{m'} \to \mathbb{R}^{m'}$ be a continuously differentiable function with $f(\boldsymbol{a}, \boldsymbol{b}) = \boldsymbol{0}$. If its Jacobian matrix $J_{f_i(\boldsymbol{x},\boldsymbol{y}),\boldsymbol{y}_j} = \frac{\partial f_i}{\partial \boldsymbol{y}_j}(\boldsymbol{a}, \boldsymbol{b})$ is invertible, then there exists an open set $\boldsymbol{a} \in U \subset \mathbb{R}^{n'}$ on which there exists a unique continuously differentiable function $g : U \to \mathbb{R}^{m'}$ with $g(\boldsymbol{a}) = \boldsymbol{b}$ and $\forall \boldsymbol{x} \in U : f(\boldsymbol{x}, g(\boldsymbol{x})) = 0$. Moreover,*

$$\frac{\partial g_i}{\partial \boldsymbol{x}_j}(\boldsymbol{x}) = -\left( J_{f(\boldsymbol{x},\boldsymbol{y}),\boldsymbol{y}}(\boldsymbol{x}, g(\boldsymbol{x})) \right)_{i,:}^{-1} \left( \frac{\partial}{\partial \boldsymbol{x}} f(\boldsymbol{x}, g(\boldsymbol{x})) \right)_{:,j} \tag{45}$$

Next we derive the transport plan derivatives. To apply the implicit function theorem we identify $\boldsymbol{x} = \boldsymbol{C}$ and $\boldsymbol{y} = \boldsymbol{P}$ as flattened matrices. We will index these flat matrices via index pairs to simplify interpretation. We furthermore identify

$$f(\boldsymbol{C}, \boldsymbol{P}) = \frac{\partial}{\partial \bar{\boldsymbol{P}}} \left( \langle \bar{\boldsymbol{P}}, \boldsymbol{C} \rangle_{\mathrm{F}} - \lambda H(\bar{\boldsymbol{P}}) \right) = \boldsymbol{C} + \lambda (\log \bar{\boldsymbol{P}} + 1). \tag{46}$$

The minimizer $\bar{\boldsymbol{P}}$ cannot lie on the boundary of $\boldsymbol{P}$'s valid region since $\lim_{p \to 0} \frac{\partial}{\partial p} p \log p = \lim_{p \to 0} \log p + 1 = -\infty$ and therefore $\bar{\boldsymbol{P}}_{ij} > 0$. Hence, $f(\boldsymbol{C}, \bar{\boldsymbol{P}}) = 0$ with $\bar{\boldsymbol{P}}(\boldsymbol{C}) = \arg\min_{\boldsymbol{P}} \langle \boldsymbol{P}, \boldsymbol{C} \rangle_{\mathrm{F}} - \lambda H(\boldsymbol{P})$ and we find that $g(\boldsymbol{x}) = \bar{\boldsymbol{P}}(\boldsymbol{C})$. We furthermore obtain

$$J_{f_{ij}(\boldsymbol{C},\boldsymbol{P}),\boldsymbol{P}_{ab}}(\boldsymbol{C}, \bar{\boldsymbol{P}}(\boldsymbol{C})) = \frac{\partial}{\partial \bar{\boldsymbol{P}}_{ab}} \left( \boldsymbol{C}_{ij} + \lambda (\log \bar{\boldsymbol{P}}_{ij} + 1) \right) = \delta_{ia} \delta_{jb} \lambda / \bar{\boldsymbol{P}}_{ij}, \tag{47}$$

$$\frac{\partial}{\partial \boldsymbol{C}_{kl}} f_{ab}(\boldsymbol{C}, \bar{\boldsymbol{P}}(\boldsymbol{C})) = \frac{\partial}{\partial \boldsymbol{C}_{kl}} \left( \boldsymbol{C}_{ab} + \lambda (\log \bar{\boldsymbol{P}}_{ab} + 1) \right) = \delta_{ak} \delta_{bl}. \tag{48}$$

$J_{f_{ij}(\boldsymbol{C},\boldsymbol{P}),\boldsymbol{P}_{ab}}(\boldsymbol{C}, \bar{\boldsymbol{P}}(\boldsymbol{C}))$ is hence a diagonal matrix and invertible since $\lambda / \bar{\boldsymbol{P}}_{ij} > 0$. We can thus use the implicit function theorem and obtain

$$\begin{aligned}
\frac{\partial \bar{\boldsymbol{P}}_{ij}}{\partial \boldsymbol{C}_{kl}} &= -\sum_{a,b} \left( J_{f_{ij}(\boldsymbol{C},\boldsymbol{P}),\boldsymbol{P}_{ab}}(\boldsymbol{C}, \bar{\boldsymbol{P}}(\boldsymbol{C})) \right)^{-1} \frac{\partial}{\partial \boldsymbol{C}_{kl}} f_{ab}(\boldsymbol{C}, \bar{\boldsymbol{P}}(\boldsymbol{C})) \\
&= -\sum_{a,b} \delta_{ia} \delta_{jb} \frac{1}{\lambda} \bar{\boldsymbol{P}}_{ij} \delta_{ak} \delta_{bl} = -\frac{1}{\lambda} \bar{\boldsymbol{P}}_{ij} \delta_{ik} \delta_{jl}.
\end{aligned} \tag{49}$$

To extend this result to LCN-OT we use

$$\frac{\partial \bar{\boldsymbol{P}}_{U,ij}}{\partial \bar{\boldsymbol{P}}_{\mathrm{Nys},ab}} = \frac{\partial}{\partial \bar{\boldsymbol{P}}_{\mathrm{Nys},ab}} \sum_k \bar{\boldsymbol{P}}_{\mathrm{Nys},ik} \bar{\boldsymbol{P}}_{W,kj}^\dagger = \delta_{ia} \bar{\boldsymbol{P}}_{W,bj}^\dagger, \tag{50}$$

$$\frac{\partial \bar{\boldsymbol{P}}_{W,ij}}{\partial \bar{\boldsymbol{P}}_{\mathrm{Nys},ab}} = \frac{\partial}{\partial \bar{\boldsymbol{P}}_{\mathrm{Nys},ab}} \sum_k \bar{\boldsymbol{P}}_{U,ik}^\dagger \bar{\boldsymbol{P}}_{\mathrm{Nys},kj} = \delta_{jb} \bar{\boldsymbol{P}}_{U,ia}^\dagger \tag{51}$$

and the chain rule to obtain

$$
\begin{aligned}
\frac{\partial \bar{P}_{U,ij}}{\partial U_{kl}} &= \sum_{a,b,c,d} \frac{\partial C_{\mathrm{Nys},cd}}{\partial U_{kl}} \frac{\partial \bar{P}_{\mathrm{Nys},ab}}{\partial C_{\mathrm{Nys},cd}} \frac{\partial \bar{P}_{U,ij}}{\partial \bar{P}_{\mathrm{Nys},ab}} \\
&= \sum_{a,b,c,d} \left(-\lambda \delta_{ck} \frac{W_{ld}}{K_{\mathrm{Nys},cd}}\right)\left(-\frac{1}{\lambda}\bar{P}_{\mathrm{Nys},ab}\delta_{ac}\delta_{bd}\right)\left(\delta_{ia}\bar{P}^{\dagger}_{W,bj}\right) \\
&= \sum_{b} W_{lb} \frac{\bar{P}_{\mathrm{Nys},ib}}{K_{\mathrm{Nys},ib}}\bar{P}^{\dagger}_{W,bj}\delta_{ik} = \delta_{ik}\sum_{b} s_i W_{lb} t_b \bar{P}^{\dagger}_{W,bj} = \delta_{ik} s_i \sum_{b} \bar{P}_{W,lb}\bar{P}^{\dagger}_{W,bj} \\
&= \delta_{ik}\delta_{jl}s_i,
\end{aligned}
\tag{52}
$$

$$
\begin{aligned}
\frac{\partial \bar{P}_{W,ij}}{\partial U_{kl}} &= \sum_{a,b,c,d} \frac{\partial C_{\mathrm{Nys},cd}}{\partial U_{kl}} \frac{\partial \bar{P}_{\mathrm{Nys},ab}}{\partial C_{\mathrm{Nys},cd}} \frac{\partial \bar{P}_{W,ij}}{\partial \bar{P}_{\mathrm{Nys},ab}} \\
&= \sum_{a,b,c,d} \left(-\lambda \delta_{ck} \frac{W_{ld}}{K_{\mathrm{Nys},cd}}\right)\left(-\frac{1}{\lambda}\bar{P}_{\mathrm{Nys},ab}\delta_{ac}\delta_{bd}\right)\left(\delta_{jb}\bar{P}^{\dagger}_{U,ia}\right) \\
&= W_{lj} \frac{\bar{P}_{\mathrm{Nys},kj}}{K_{\mathrm{Nys},kj}}\bar{P}^{\dagger}_{U,ik} = W_{lj} t_j \bar{P}^{\dagger}_{U,ik} s_k = \bar{P}^{\dagger}_{U,ik} s_k \bar{P}_{W,lj},
\end{aligned}
\tag{53}
$$

$$
\begin{aligned}
\frac{\partial \bar{P}_{U,ij}}{\partial W_{kl}} &= \sum_{a,b,c,d} \frac{\partial C_{\mathrm{Nys},cd}}{\partial W_{kl}} \frac{\partial \bar{P}_{\mathrm{Nys},ab}}{\partial C_{\mathrm{Nys},cd}} \frac{\partial \bar{P}_{U,ij}}{\partial \bar{P}_{\mathrm{Nys},ab}} \\
&= \sum_{a,b,c,d} \left(-\lambda \delta_{dl} \frac{U_{ck}}{K_{\mathrm{Nys},cd}}\right)\left(-\frac{1}{\lambda}\bar{P}_{\mathrm{Nys},ab}\delta_{ac}\delta_{bd}\right)\left(\delta_{ia}\bar{P}^{\dagger}_{W,bj}\right) \\
&= U_{ik} \frac{\bar{P}_{\mathrm{Nys},il}}{K_{\mathrm{Nys},il}}\bar{P}^{\dagger}_{W,lj} = s_i U_{ik} t_l \bar{P}^{\dagger}_{W,lj} = \bar{P}_{U,ik} t_l \bar{P}^{\dagger}_{W,lj},
\end{aligned}
\tag{54}
$$

$$
\begin{aligned}
\frac{\partial \bar{P}_{W,ij}}{\partial W_{kl}} &= \sum_{a,b,c,d} \frac{\partial C_{\mathrm{Nys},cd}}{\partial W_{kl}} \frac{\partial \bar{P}_{\mathrm{Nys},ab}}{\partial C_{\mathrm{Nys},cd}} \frac{\partial \bar{P}_{W,ij}}{\partial \bar{P}_{\mathrm{Nys},ab}} \\
&= \sum_{a,b,c,d} \left(-\lambda \delta_{dl} \frac{U_{ck}}{K_{\mathrm{Nys},cd}}\right)\left(-\frac{1}{\lambda}\bar{P}_{\mathrm{Nys},ab}\delta_{ac}\delta_{bd}\right)\left(\delta_{jb}\bar{P}^{\dagger}_{U,ia}\right) \\
&= \sum_{a} U_{ak} \frac{\bar{P}_{\mathrm{Nys},aj}}{K_{\mathrm{Nys},aj}}\bar{P}^{\dagger}_{U,ia}\delta_{jl} = \delta_{jl}\sum_{a} s_a U_{ak} t_j \bar{P}^{\dagger}_{U,ia} = \delta_{jl} t_j \sum_{a} \bar{P}^{\dagger}_{U,ia}\bar{P}_{U,ak} \\
&= \delta_{ik}\delta_{jl}t_j.
\end{aligned}
\tag{55}
$$

$\frac{\partial \bar{P}^{\mathrm{sp}}}{\partial \log K^{\mathrm{sp}}}$ and $\frac{\partial \bar{P}^{\mathrm{sp}}_{\mathrm{Nys}}}{\partial \log K^{\mathrm{sp}}_{\mathrm{Nys}}}$ directly follow from $\frac{\partial \bar{P}}{\partial C}$.

We can calculate the pseudoinverses $\bar{P}^{\dagger}_{U} = (\bar{P}^{T}_{U}\bar{P}_{U})^{-1}\bar{P}_{U}$ and $\bar{P}^{\dagger}_{W} = \bar{P}^{T}_{W}(\bar{P}_{W}\bar{P}^{T}_{W})^{-1}$ in time $\mathcal{O}((n+m)l^2 + l^3)$ since $\bar{P}_{U} \in \mathbb{R}^{n \times l}$ and $\bar{P}_{W} \in \mathbb{R}^{l \times m}$. We do not fully instantiate the matrices required for backpropagation but instead save their decompositions, similar to the transport plan $\bar{P}_{\mathrm{Nys}} = \bar{P}_{U}\bar{P}_{W}$. We can then compute backpropagation in time $\mathcal{O}((n+m)l^2)$ by applying the sums over $i$ and $j$ in the right order. We thus obtain $\mathcal{O}((n+m)l^2 + l^3)$ overall runtime for backpropagation. $\square$

## H    Choosing LSH Neighbors and Nyström Landmarks

We focus on two LSH methods for obtaining near neighbors. Cross-polytope LSH (Andoni et al., 2015) uses a random projection matrix $R \in \mathbb{R}^{d \times b/2}$ with the number of hash buckets $b$, and then decides on the hash bucket via $h(x) = \arg\max([x^T R \,\|\, -x^T R])$, where $\|$ denotes concatenation. K-means LSH computes k-means and uses the clusters as hash buckets.

We further improve the sampling probabilities of cross-polytope LSH via the AND-OR construction. In this scheme we calculate $B \cdot r$ hash functions, divided into $B$ sets (hash bands) of $r$ hash functions

Table 5: Graph dataset statistics.

| | Graph type | Distance | Distance (test set) | | Graphs train/val/test | Avg. nodes per graph | Avg. edges per graph | Node types | Edge types |
|---|---|---|---|---|---|---|---|---|---|
| | | | Mean | Std. dev. | | | | | |
| AIDS30 | Molecules | GED | 50.5 | 16.2 | 144/48/48 | 20.6 | 44.6 | 53 | 4 |
| Linux | Program dependence | GED | 0.567 | 0.181 | 600/200/200 | 7.6 | 6.9 | 7 | - |
| Pref. att. | Initial attractiveness | GED | 106.7 | 48.3 | 144/48/48 | 20.6 | 75.4 | 6 | 4 |
| Pref. att. 200 | Initial attractiveness | PM | 0.400 | 0.102 | 144/48/48 | 199.3 | 938.8 | 6 | - |
| Pref. att. 2k | Initial attractiveness | PM | 0.359 | 0.163 | 144/48/48 | 2045.6 | 11330 | 6 | - |
| Pref. att. 20k | Initial attractiveness | PM | 0.363 | 0.151 | 144/48/48 | 20441 | 90412 | 6 | - |

each. A pair of points is considered as neighbors if any hash band matches completely. K-means LSH does not work well with the AND-OR construction since its samples are highly correlated. For large datasets we use hierarchical k-means instead (Paulevé et al., 2010; Nistér & Stewénius, 2006).

Since the graph transport network (GTN) uses the $L_2$ distance between embeddings as a cost function we use (hierarchical) k-means LSH and k-means Nyström in both sparse OT and LCN-OT. For embedding alignment we use cross-polytope LSH for sparse OT since similarities are measured via the dot product. For LCN-OT we found that using k-means LSH works better with Nyström using k-means++ sampling than cross-polytope LSH. This is most likely due to a better alignment between LSH samples and Nyström. We convert the cosine similarity to a distance via $d_{\cos} = \sqrt{1 - \frac{x_p^T x_q}{\|x_p\|_2 \|x_q\|_2}}$ (Berg et al., 1984) to use k-means with dot product similarity. Note that this is actually based on cosine similarity, not the dot product. Due to the balanced nature of OT we found this more sensible than maximum inner product search (MIPS). For both experiments we also experimented with uniform and recursive RLS sampling but found that the above mentioned methods work better.

## I  IMPLEMENTATIONAL DETAILS

Our implementation runs in batches on a GPU via PyTorch (Paszke et al., 2019) and PyTorch Scatter (Fey & Lenssen, 2019). To avoid over- and underflows we use log-stabilization throughout, i.e. we save all values in log-space and compute all matrix-vector products and additions via the log-sum-exp trick $\log \sum_i e^{x_i} = \max_j x_j + \log(\sum_i e^{x_i - \max_j x_j})$. Since the matrix $A$ is small we compute its inverse using double precision to improve stability. Surprisingly, we did not observe any benefit from using the Cholesky decomposition or not calculating $A^{-1}$ and instead solving the equation $B = AX$ for $X$. We furthermore precompute $W = A^{-1}V$ to avoid unnecessary operations.

We use 3 layers and an embedding size $H_N = 32$ for GTN. The MLPs use a single hidden layer, biases and LeakyReLU non-linearities. The single-head MLP uses an output size of $H_{N, \text{match}} = H_N$ and a hidden embedding size of $4H_N$, i.e. the same as the concatenated node embedding, and the multi-head MLP uses a hidden embedding size of $H_N$. To stabilize initial training we scale the node embeddings by $\frac{\bar{d}}{\bar{n}\sqrt{H_{N, \text{match}}}}$ directly before calculating OT. $\bar{d}$ denotes the average graph distance in the training set, $\bar{n}$ the average number of nodes per graph, and $H_{N, \text{match}}$ the matching embedding size, i.e. 32 for single-head and 128 for multi-head OT.

## J  GRAPH DATASET GENERATION AND EXPERIMENTAL DETAILS

The dataset statistics are summarized in Table 5. Each dataset contains the distances between all graph pairs in each split, i.e. 10 296 and 1128 distances for preferential attachment. The AIDS dataset was generated by randomly sampling graphs with at most 30 nodes from the original AIDS dataset (Riesen & Bunke, 2008). Since not all node types are present in the training set and our choice of GED is permutation-invariant w.r.t. types, we permuted the node types so that there are no previously unseen types in the validation and test sets. For the preferential attachment datasets we first generated 12, 4, and 4 undirected "seed" graphs (for train, val, and test) via the initial attractiveness model with randomly chosen parameters: 1 to 5 initial nodes, initial attractiveness of 0 to 4 and $1/2\bar{n}$ and $3/2\bar{n}$ total nodes, where $\bar{n}$ is the average number of nodes (20, 200, 2000, and 20 000). We then randomly label every node (and edge) in these graphs uniformly. To obtain the remaining graphs we edit the "seed" graphs between $\bar{n}/40$ and $\bar{n}/20$ times by randomly adding, type editing, or removing nodes

Table 6: Hyperparameters for the Linux dataset.

| | lr | batchsize | layers | emb. size | $L_2$ reg. | $\lambda_{\text{base}}$ |
|---|---|---|---|---|---|---|
| SiamMPNN | $1 \times 10^{-4}$ | 256 | 3 | 32 | $5 \times 10^{-4}$ | - |
| GMN | $1 \times 10^{-4}$ | 20 | 3 | 64 | 0 | - |
| GTN, 1 head | 0.01 | 1000 | 3 | 32 | $1 \times 10^{-6}$ | 1.0 |
| 8 OT heads | 0.01 | 1000 | 3 | 32 | $1 \times 10^{-6}$ | 1.0 |
| Balanced OT | 0.01 | 1000 | 3 | 32 | $1 \times 10^{-6}$ | 2.0 |

Table 7: Hyperparameters for the AIDS dataset.

| | lr | batchsize | layers | emb. size | $L_2$ reg. | $\lambda_{\text{base}}$ |
|---|---|---|---|---|---|---|
| SiamMPNN | $1 \times 10^{-4}$ | 256 | 3 | 32 | $5 \times 10^{-4}$ | - |
| SimGNN | $1 \times 10^{-3}$ | 1 | 3 | 32 | 0.01 | - |
| GMN | $1 \times 10^{-2}$ | 128 | 3 | 32 | 0 | - |
| GTN, 1 head | 0.01 | 100 | 3 | 32 | $5 \times 10^{-3}$ | 0.1 |
| 8 OT heads | 0.01 | 100 | 3 | 32 | $5 \times 10^{-3}$ | 0.075 |
| Balanced OT | 0.01 | 100 | 3 | 32 | $5 \times 10^{-3}$ | 0.1 |
| Nyström | 0.015 | 100 | 3 | 32 | $5 \times 10^{-3}$ | 0.2 |
| Multiscale | 0.015 | 100 | 3 | 32 | $5 \times 10^{-3}$ | 0.2 |
| Sparse OT | 0.015 | 100 | 3 | 32 | $5 \times 10^{-3}$ | 0.2 |
| LCN-OT | 0.015 | 100 | 3 | 32 | $5 \times 10^{-3}$ | 0.2 |

and edges. Editing nodes and edges is 4x and adding/deleting edges 3x as likely as adding/deleting nodes. Most of these numbers were chosen arbitrarily, aiming to achieve a somewhat reasonable dataset and process. We found that the process of first generating seed graphs and subsequently editing these is crucial for obtaining meaningfully structured data to learn from. For the GED we choose an edit cost of 1 for changing a node or edge type and 2 for adding or deleting a node or an edge.

We represent node and edge types as one-hot vectors. We train all models except SiamMPNN (which uses SGD) and GTN on Linux with the Adam optimizer and mean squared error (MSE) loss for up to 300 epochs and reduce the learning rate by a factor of 10 every 100 steps. On Linux we train for up to 1000 epochs and reduce the learning rate by a factor of 2 every 100 steps. We use the parameters from the best epoch based on the validation set. We choose hyperparameters for all models using multiple steps of grid search on the validation set, see Tables 6 to 8 for the final values. We use the originally published result of SimGNN on Linux and thus don't provide its hyperparameters. GTN uses 500 Sinkhorn iterations. We obtain the final entropy regularization parameter from $\lambda_{\text{base}}$ via $\lambda = \lambda_{\text{base}} \frac{\bar{d}}{\bar{n}} \frac{1}{\log n}$, where $\bar{d}$ denotes the average graph distance and $\bar{n}$ the average number of nodes per graph in the training set. The factor $\bar{d}/\bar{n}$ serves to estimate the embedding distance scale and $1/\log n$ counteracts the entropy scaling with $n \log n$. Note that the entropy regularization parameter was small, but always far from 0, which shows that entropy regularization actually has a positive

Table 8: Hyperparameters for the preferential attachment GED dataset.

| | lr | batchsize | layers | emb. size | $L_2$ reg. | $\lambda_{\text{base}}$ |
|---|---|---|---|---|---|---|
| SiamMPNN | $1 \times 10^{-4}$ | 256 | 3 | 64 | $1 \times 10^{-3}$ | - |
| SimGNN | $1 \times 10^{-3}$ | 4 | 3 | 32 | 0 | - |
| GMN | $1 \times 10^{-4}$ | 20 | 3 | 64 | 0 | - |
| GTN, 1 head | 0.01 | 100 | 3 | 32 | $5 \times 10^{-4}$ | 0.2 |
| 8 OT heads | 0.01 | 100 | 3 | 32 | $5 \times 10^{-3}$ | 0.075 |
| Balanced OT | 0.01 | 100 | 3 | 32 | $5 \times 10^{-4}$ | 0.2 |
| Nyström | 0.02 | 100 | 3 | 32 | $5 \times 10^{-5}$ | 0.2 |
| Multiscale | 0.02 | 100 | 3 | 32 | $5 \times 10^{-5}$ | 0.2 |
| Sparse OT | 0.02 | 100 | 3 | 32 | $5 \times 10^{-5}$ | 0.2 |
| LCN-OT | 0.02 | 100 | 3 | 32 | $5 \times 10^{-5}$ | 0.2 |

Table 9: Runtimes (ms) of Sinkhorn approximations for EN-DE embeddings at different dataset sizes. Full Sinkhorn scales quadratically, while all approximationes scale at most linearly with the size. Sparse approximations are 2-4x faster than low-rank approximations, and factored OT is multiple times slower due to its iterative refinement scheme. Note that similarity matrix computation time ($K$) primarily depends on the LSH/Nyström method, not the OT approximation.

|  | $N = 10000$ | | $N = 20000$ | | $N = 50000$ | |
|---|---|---|---|---|---|---|
|  | $K$ | OT | $K$ | OT | $K$ | OT |
| Full Sinkhorn | 8 | 2950 | 29 | 11 760 | OOM | OOM |
| Factored OT | 29 | 809 | 32 | 1016 | 55 | 3673 |
| Multiscale OT | 90 | 48 | 193 | 61 | 521 | 126 |
| Nyström Skh. | 29 | 135 | 41 | 281 | 79 | 683 |
| Sparse Skh. | 42 | 46 | 84 | 68 | 220 | 137 |
| LCN-Sinkhorn | 101 | 116 | 242 | 205 | 642 | 624 |

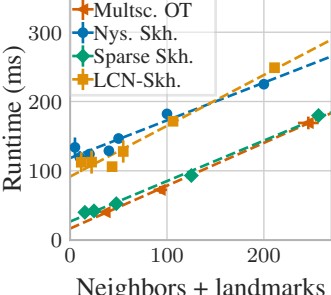

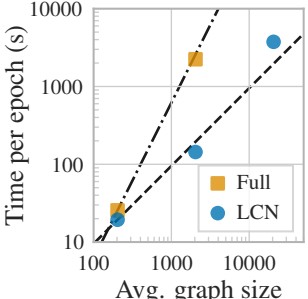

Figure 5: Runtime scales linearly with the number of neighbors/landmarks for all relevant Sinkhorn approximation methods.

Figure 6: Log-log runtime per epoch for GTN with full Sinkhorn and LCN-Sinkhorn. LCN-Sinkhorn scales almost linearly with graph size while sustaining similar accuracy.

effect on learning. On the pref. att. 200 dataset we use no $L_2$ regularization, $\lambda_{\text{base}} = 0.5$, and a batch size of 200. For pref. att. 2k we use $\lambda_{\text{base}} = 2$ and a batch size of 20 for full Sinkhorn and 100 for LCN-OT. For pref. att. 20k we use $\lambda_{\text{base}} = 50$ and a batch size of 4. $\lambda_{\text{base}}$ scales with graph size due to normalization of the PM kernel.

For LCN-OT we use roughly 10 neighbors for LSH (20 k-means clusters) and 10 k-means landmarks for Nyström on pref. att. 200. We double these numbers for pure Nyström Sinkhorn, sparse OT, and multiscale OT. For pref. att. 2k we use around 15 neighbors ($10 \cdot 20$ hierarchical clusters) and 15 landmarks and for pref. att. 20k we use roughly 30 neighbors ($10 \cdot 10 \cdot 10$ hierarchical clusters) and 20 landmarks. The number of neighbors for the 20k dataset is higher and strongly varies per iteration due to the unbalanced nature of hierarchical k-means. This increase in neighbors and landmarks and PyTorch's missing support for ragged tensors largely explains LCN-OT's deviation from perfectly linear runtime scaling.

We perform all runtime measurements on a compute node using one Nvidia GeForce GTX 1080 Ti, two Intel Xeon E5-2630 v4, and 256GB RAM.

## K RUNTIMES

Table 9 compares the runtime of the full Sinkhorn distance with different approximation methods using 40 neighbors/landmarks. We separate the computation of approximate $K$ from the optimal transport computation (Sinkhorn iterations), since the former primarily depends on the LSH and Nyström methods we choose. We observe a 2-4x speed difference between sparse (multiscale OT and sparse Sinkhorn) and low-rank approximations (Nyström Sinkhorn and LCN-Sinkhorn), while factored OT is multiple times slower due to its iterative refinement scheme. In Fig. 5 we observe that this runtime gap stays constant independent of the number of neighbors/landmarks, i.e. the relative

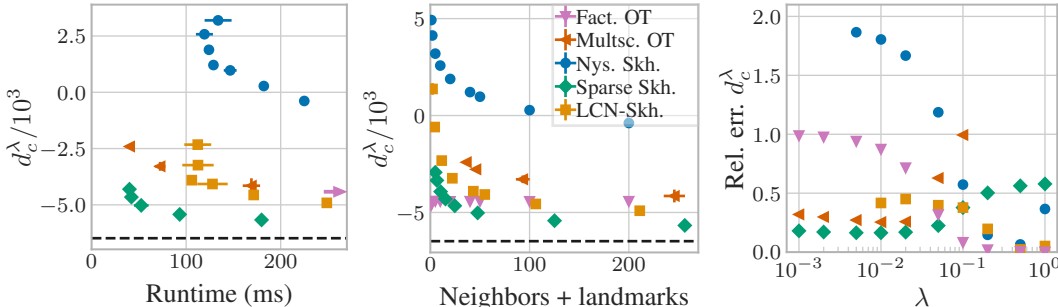

Figure 7: Sinkhorn distance approximation at different runtimes (varied via the number of neighbors/landmarks). The dashed line denotes the true Sinkhorn distance. Sparse Sinkhorn consistently performs best. The arrow indicates factored OT results far outside the depicted range.

Figure 8: Sinkhorn distance approximation for varying computational budget. The dashed line denotes the true Sinkhorn distance. Sparse Sinkhorn mostly performs best, with LCN-Sinkhorn coming in second. Factored OT performs well with very few landmarks.

Figure 9: Sinkhorn distance approximation for varying entropy regularization $\lambda$ at constant computational budget. Sparse Sinkhorn performs best for low $\lambda$, LCN-Sinkhorn for moderate and high $\lambda$ and factored OT for high $\lambda$.

difference decreases as we increase the number of neighbors/landmarks. This gap could either be due to details in low-level CUDA implementations and hardware or the fact that low-rank approximations require 2x as many multiplications for the same number of neighbors/landmarks. In either case, both Table 9 and Fig. 5 show that the runtimes of all approximations scale linearly both in the dataset size and the number of neighbors and landmarks, while full Sinkhorn scales quadratically.

We furthermore investigate whether GTN with approximate Sinkhorn indeed scales log-linearly with the graph size by generating preferential attachment graphs with 200, 2000, and 20 000 nodes ($\pm 50\,\%$). We use the Pyramid matching (PM) kernel (Nikolentzos et al., 2017) as prediction target. Fig. 6 shows that the runtime of LCN-Sinkhorn scales almost linearly (dashed line) and regular full Sinkhorn quadraticly (dash-dotted line) with the number of nodes, despite both achieving similar accuracy and LCN using slightly more neighbors and landmarks on larger graphs to sustain good accuracy. Full Sinkhorn went out of memory for the largest graphs.

## L    DISTANCE APPROXIMATION

Figs. 7 and 8 show that for the chosen $\lambda = 0.05$ sparse Sinkhorn offers the best trade-off between computational budget and distance approximation, with LCN-Sinkhorn and multiscale OT coming in second. Factored OT is again multiple times slower than the other methods and thus not included in Fig. 7. Note that $d_c^\lambda$ can be negative due to the entropy offset. This picture changes as we increase the regularization. For higher regularizations LCN-Sinkhorn is the most precise at constant computational budget (number of neighbors/landmarks), as shown in Fig. 9. Note that the crossover points in this figure roughly coincide with those in Fig. 4. Keep in mind that in most cases the OT plan is more important than the raw distance approximation, since it determines the training gradient and tasks like embedding alignment don't use the distance at all. This becomes evident in the fact that sparse Sinkhorn achieves a better distance approximation than LCN-Sinkhorn but performs worse in both downstream tasks investigated in Sec. 7.

