# OpenReview forum: "Warpspeed Computation of Optimal Transport, Graph Distances, and Embedding Alignment"
_ICLR.cc/2021/Conference — Reject_

### Official Review · AnonReviewer1 · 2020-10-26
**Interesting paper and potentially significant.But theory is lacking, and experimental section can be improved**

**Rating:** 6
**Confidence:** 4

**Review:**

Overall, I found this paper interesting, and I think it does address a relevant problem for the community.


I have been playing with Nystrom approximations myself and I know the results are a bit disappointing, but it is grounded on strong theory. Then, it is pretty much welcome that attempts to patch the problems of Nystroms are made, so Optimal Transport becomes more scalable.

The reason for why my judgement is below acceptance is that I believe both theory and results altogether are not strong enough so they live up to what is promised in the abstract. Typically when new methods are proposed with the promise of bridging a gap and solving a relevant problem, I hope they will have a thorough theoretical justification, or the results will be compelling. None of this convincing enough here.

1)On the theoretical side, I missed a convergence analysis, as the one in the Nystrom method. The theory side focuses on some derivative calculations, but I would love to see how the interplay between sparsity, Nystrom method and LSH lead to better convergence. I acknowledge this can be hard to do, though. Without having the theory it is hard to understand whether any reported experimental result is a consequence of choosing particularly good example for their method.

2)I found the experimental/methodological side was a bit disconnected from the rest; the paper contains  several vignettes about some applications/improvements in the practical side, but when reading that felt like belonging to other paper. I recommend authors work in creating a more coherent story
2a) I found the discussion on Multi-Head OT unjustified and even a bit misleading. The Authors refer to some NLP papers, like arguing OT plays a role there. But none of these papers have any OT at all. The analogy of  "softmax for rows" is not convincing as this is simply a softmax applied many times. There is a world of difference between that and the result of sinkhorn algorithm, but the narrative seems to downplay the actual difference between them. I recommend the authors elaborate more on this connection, because otherwise it is hard to follow (and the subsequent results).
2b)Results on translation seem impressive (Table 2), but raise a concern. Why would your method outperform Sinkhorn if it is only approximation? is it perhaps the result of randomness? since an explanation of this phenomenon is missing I am led to believing. Authors should improve the exposition of the baseline "original". Why does full Sinkhorn does better than original?. In summary, I think authors should improve the discussion about the validity/significance of their empirical results, highlighting the regimes when they are supposed to express and when they are not.
2c)The main figure is Fig 2. I recommend authors build on this and expand those results so it is clear when their method is better and when it is not

---

> ### Author Response · Authors · 2020-11-21
> **New theoretical convergence analysis, streamlined text**
>
> ## Ad 1) New theoretical convergence analysis
> As suggested, we have now added a full convergence analysis to the paper (see Section 4). In our understanding, such an analysis requires 2 parts:
> 1. How good is the approximation?
> 2. How fast does the approximation converge?
>
> To answer (1) we first analyze the maximum error of the similarity matrix computed by LCN in comparison to regular Nyström in (a) a uniformly distributed data model and (b) a clustered data model. We then use the result by Altschuler et al. 2019 to relate these bounds to bounds for the final Sinkhorn approximation.
> To answer (2) we only need to slightly modify the bound by Dvurechensky et al. 2018 to show that both sparse Sinkhorn and LCN-Sinkhorn have the same convergence rate bound as regular Sinkhorn.
>
> ## Ad 2) Streamlined text
> We tried to make the paper more coherent by improving the introduction and integrating the section on enhanced OT in the graph transport network section, for which these are an important component.
>
> Ad 2a) We agree and have mostly removed this part of the motivation.
>
> Ad 2b) LCN-Sinkhorn performing better than full Sinkhorn on word embedding alignment is most likely due to regularization effects caused by the approximation, which lead to better generalization. The original method used by Grave et al. 2019 performs Sinkhorn on an iteratively increasing, randomly sampled subset of the embeddings. Our experiments clearly show that this approach is far from ideal. We have added these explanations to the paper.
>
> _All_ figures and tables in the paper show standard deviations, thus the significance of our results is clearly marked. Furthermore, please note that as opposed to previous works we don’t just show results on some potentially cherry-picked datasets but make the effort of fully integrating our approximations in a pipeline and evaluate end-to-end real-world performance.
>
> Ad 2c) We have restructured the text to make this figure’s description more prominent.

---

> ### Author Response · Authors · 2020-11-24
> **Gentle Reminder to Revisit the Improved Paper**
>
> Seeing that the end of the discussion period is drawing near, we would like to again highlight the significant improvements your comments prompted us to make to the paper.
>
> Considering the significant effort we have put into deriving the **theoretical convergence analysis** you were missing in the original version, we sincerely hope that you can find the time to revisit the substantially improved revised version. Other reviewers have already positively highlighted the added theoretical foundation provided by this analysis.

---

### Official Review · AnonReviewer3 · 2020-10-27
**Good practical approach, but not sufficiently clear for a general audiece**

**Rating:** 7
**Confidence:** 4

**Review:**

Summary:
The paper considers the problem of approximating Sinkhorn divergence and corresponding transportation plan by combining low-rank and sparse approximation for the Sinkhorn kernel and using Nystrom iterations as a substitute for Sinkhorn's iterations. The corresponding approach is amenable to differentiation and can be used as a building block in different architectures. Numerical experiments in several settings are performed to compare  the proposed approach with existing ones and demonstrate its scalability.

Evaluation:
I believe the proposed framework is a valuable contribution in terms of practical performance and wide list of applications where OT could not be used before because of the high computational cost. So, I would recommend accepting this paper.

Pros:
1. High scalability of the proposed approach and linear up to log factors in dimension complexity.
2. Flexibility of the framework due to a combination of sparse and low-rank approximations, which are complementary to each other.

Cons:
1. Some parts of the paper seem to be not clear for a general audience.

a. First page. $n$ is undefined.

b. First paragraph of Sect. 2. What is "set of embeddings"?

c. Last but one paragraph on p.2. $d$ is not defined.

d. In (1) $F$ stands for the Frobenius product, does it?

e. Proposition 1. $N$ is not defined.

f. First paragraph of Sect. 5. What is "OT with multiple heads"?

g. What is meant as embedding?

h. In the experiments, what is used as the cost function to define the Sinkhorn kernel $K$? If this is an $L_2$ distance, the convolutions can be used to accelerate the standard Sinkhorn and it would be nice to see the comparisons with convolutional Sinkhorn, which is also log linear.

i. Appendix A. $B,r,b$ are not defined when they are first used.

j. In (17), how was the last equality obtained?

k. In (19), (20), how were the first equalities obtained?

l. Appendix E, first paragraph. What is "log-sum-exp trick"?

m. Appendix G. What is "similarity matrix"?

2. As far as I understood, the proposed approach is not amenable to parallel computations on GPU as opposed to standard Sinkhorn.


Minor comments
1. Maybe it is too strong to state in the abstract that this is the first log-linear time algorithm given that when the Sinkhorn kernel corresponds to a convolution, the Sinkhorn's algorithm is log-linear by using the FFT.
2. Bibliographical note. (Altschuler et al., 2017) did not show $1/\varepsilon^2$ bound for the Greenkhorn. Their bound for Greenkhorn is the same $1/\varepsilon^3$ as for the Sinkhorn. The bound for Sinkhorn was improved to $1/\varepsilon^2$ in http://proceedings.mlr.press/v80/dvurechensky18a.html and the bound for Greenkhorn was improved to $1/\varepsilon^2$ in http://proceedings.mlr.press/v97/lin19a.html.
3. Bibliographical note. Quadratic regularization for OT was proposed in https://arxiv.org/abs/1704.08200.
4. Appendix A. I believe that in this framework a general value of the regularization parameter $\lambda$ is used. If it is the case, then the number of Sinkhorn iterations to find an $\varepsilon$-solution to the regularized problem is $1/(\varepsilon \lambda)$. This follows from http://proceedings.mlr.press/v80/dvurechensky18a.html Theorem 1 and an estimate for $R$ in Lemma 1. The bound $1/\varepsilon^2$ corresponds to finding an $\varepsilon$-solution for the non-regularized problem. In this case one has to set $\lambda=\varepsilon/(4 \ln n)$, which may be too small.

---

> ### Author Response · Authors · 2020-11-21
> **Fully parallel, improved readability and definitions**
>
> ## Improved readability
> We have incorporated all of your suggestions and clarified all your questions in the revised version of the paper (including your minor comments). We have furthermore improved the introduction, pushed back the related work section, and integrated the enhanced OT section in the graph transport section to make the paper easier to understand.
>
> Regarding your particular comments:
> - We have added definitions and explanations to address your Cons 1a-g, i-m
> - Ad h1) The cost functions are described in Appendix H. We use the negative dot-product for word embeddings (experiments 1 & 2) and the L2 distance in GTN (experiment 3).
> - Ad h2 & comment 1) There are many special cases for which log-linear algorithms exist and we tried to cover them in our related work section. The convolutional approach you mentioned (we assume you refer to Solomon et al. 2015. “Convolutional wasserstein distances”) is only applicable to very low-dimensional (i.e. 2-3 dimensional) data, not the high-dimensional spaces we focus on. For example, the word embeddings we work with have 300 dimensions.
> - Ad comments 2-3) We have changed the related work accordingly.
> - Ad minor comment 4) We now include Dvurechensky’s result directly in the paper and have made the bound in Appendix A more precise. The $\lambda$ we chose in our experiments was the one that performed best for full Sinkhorn. Also, sparse Sinkhorn actually performs especially well for very low $\lambda$ (see Fig. 4), so we don’t expect this to be a problem.
>
> ## Fully parallel
> Our method is actually fully parallelizable. All experiments and runtime measurements were performed with a massively parallel PyTorch implementation running on GPUs.

---

> > ### Comment · AnonReviewer3 · 2020-11-22
> > **I would like to thank the authors for the answers and keep my score unchanged.**
> >
> > I would like to thank the authors for their answers. I believe that the quality of the paper has improved after the revision, especially the method is now more theoretically grounded. Given that my initial rating was already quite positive, I'm tending to keep my score unchanged.

---

### Official Review · AnonReviewer2 · 2020-10-29
**Review on Warpspeed Computation of Optimal Transport, Graph Distances, and Embedding Alignment**

**Rating:** 6
**Confidence:** 3

**Review:**

Proposal:
- First log-linear time algorithms for entropy-regularized OT that work for complex
  real-world tasks using high-dimensional spaces with little to no loss in accuracy
  ... many claims in one statement
- Locally Corrected Nyström
  ... this would deserve a single paper - just to proof everything is still fine and valid
  in particular to alternatives
- In the way this paper is written I am positive it gets accepted
  (because it fits the writing style of the more recent papers in the field)
  and contains sufficient novelty
  ... but I always wonder if good (published) science originates from clarity (or confusion)

comments:
- sorry but the text is very hard to follow !
- 'Optimal transport is concerned with the problem' (on p.3) - I think some introductive
  work may not harm in the first page
- the reader is thrown up by terms and references - in my view more confusing than enlighting
  --> it may not harm to add some brief explainations of terms (from Cuturi:)
  'A transport plan is a flow on that graph satisfying source (a i flowing out of each node i)
   and sink (b j flowing into each node j 0 ) ... --- which is simple an optimal flow
  in a graph ... I am not sure why we not simple can call it like this but need to come up with
  new terms
- The paper is written (following the very strange title
   ... although Cuturi did the same it would be nice if we can stop having marketing titles
  but focus a bit on science again ... in particular in 10 years many things proposed nowadays
  are not lightspeed or warpspeed anymore) like providing an all issues solving theory
  --> this does not improve the readability of the paper
  For example Eq.2 what is the 'meaning' of (s) and (t) -- I have an idea but it is not written there
- it is hard to proofread and verify a paper if it is written with the objective to confuse the
  reviewer ;-)
- widely incremental work by combing some known ideas (Nystreom, LSH, ...) - with a lot of addon
  theory which is probably correct but not very clear in the presentation
- 'Since the Nyström method is a low-rank approximation it only accounts for the
global structure of the kernel matrix K and not for the local structure around each point x.'
  - this is actually wrong (!) - if the landmarks are indepentent and the number of landmarks
  aligns with the rank of the matrix - Nystroem will provide a perfect (!) reconstruction
	--> there is a lot of work on the approximation bound of Nystroem (and related methods) - see
  e.g. work by Dhillon
- where is the definition of 'sparse approximation K^sp' used in Eq 3?
- how precisely does \bar{P} (after Eq 3) link to the part around Eq 2? - is the Kernel K_{ij} used
  here as well and / or where is the actually input data Kernel matrix
- in Eq 1 what should be a cost function here and how do you obtain C_{ij}?
- Ok Eq 4 is an actual proposal by balancing (and joining) sinkhorn and Nystroem in one distance formulation
  and it would not harm to motivate why and where you need such a distance in advance
  (problem statement --> solutions --> particular strategy --> outlined proposal --> evaluation + proofs)
- 'Most modern ML models are trained using backpropagation' - lets rephrase it as: nowadays neural network
  approaches are trained by backprop ... there are many other methods which are not at all trained by backprob
  for good reasons
- '... Usually we want to learn embeddings which act as point sets X p and X q and therefore need gradients'
  - well yes, if we stick on neural networks we need vectorial inputs and hence are often looking for (costly)
  embeddings - if we do not use NN we may not have this problem (but others)
- 'We can either estimate these gradients via automatic differentiation' - this is in general the more costly
  way to do things and I am happy to see that explicit derivations are given
- regarding table 1 --> before you come with numbers (where you measured something) it would be good to specify
  details of your scenario (which are omitted before) - in particular which data, which cost function, which parameters
  a.s.o. -- and although I understand that you like your method most it would still be good to provide some
  oldfashion baseline (and not - not from sinkhorn)
- 'We propose the graph transport network (GTN) to evaluate approximate Sinkhorn and enhanced optimal transport and advance the state of the art on this task.' --- fantastic on page 6 you actually outline a more userfriendly motivation

---

> ### Author Response · Authors · 2020-11-21
> **Improved readability and theoretical analysis**
>
> ## Contributions
> It seems to us like there has been some confusion about our paper’s contributions. Our method does not “balance (and join) Sinkhorn and Nyström in one distance”. Instead, we approximate the similarity matrix $K=\exp(-C/\lambda)$ using (1) a sparse approximation and (2) a fused sparse/Nyström approximation (which we call locally corrected Nyström). We then use this approximate K _inside_ Sinkhorn and show that doing so yields a better and faster approximation of Sinkhorn than previous methods.
>
> ## Improved readability
> We have incorporated all of your helpful suggestions in the revised version. In particular:
> - We have significantly improved the introduction, now starting with a general problem description, as suggested.
> - We added many more explanations and definitions to address your various questions, e.g. regarding $s$, $t$, $K^{\text{sp}}$, $\bar{P}_{\text{LCN}}$, and the cost function. This should prevent future misinterpretations of Eq. 5 (previously Eq. 4).
> - We improved the motivation for providing explicit gradients.
> - We have removed the OT enhancements section and moved its content to the graph transport network section, where we use them.
>
> ## Nyström approximation
> Sinkhorn uses kernels of the form $\exp(-C/\lambda)$. Kernels like these (e.g. the Gaussian kernel) typically have a reproducing kernel Hilbert space that is infinitely dimensional. The resulting Gram matrix thus always has full rank and can not be reconstructed by a low-rank matrix such as the one provided by the Nyström method (!). We have made this more clear in the paper.
>
> ## Table 1
> We are not sure if we understand your comment. The text describes the experimental setup before referring to Table 1. The goal of our method is to approximate Sinkhorn, so in this experiment it is not a baseline but the ground truth.
>
> ## Theoretical analysis
> To address your comment on validity we provide a new theoretical analysis in Section 4. In particular, we show that (a) LCN provides significant benefits over Nyström in both a uniform and clustered data model, (b) the error of approximate Sinkhorn is bounded, and (c) approximate Sinkhorn enjoys the same converge bounds as regular Sinkhorn.

---

> ### Author Response · Authors · 2020-11-24
> **Gentle Reminder to Revisit the Improved Paper**
>
> Seeing that the end of the discussion period is drawing near, we would again like to highlight that we have addressed all the issues you have raised in your review, some of which were based on an unfortunate misinterpretation.
>
> We sincerely hope that you can find the time to revisit the significantly clearer and improved new version of our paper. These improvements are to a large part due to your helpful comments and suggestions.

---

### Official Review · AnonReviewer4 · 2020-10-29
**A practical OT work that enhances performance on three application tasks**

**Rating:** 6
**Confidence:** 3

**Review:**

This paper studies how to approximate Sinkhorn computation using more efficient kernel matrix representations (low-rank approach + LSH based sparse approach). Neither of both ideas is completely new, the authors studies a combination of them that hasn't been explored in the literature, and use the proposed tech in three applications: ranking, embedding alignment, graph distance regression.

Pro:

- Authors have attempted multiple applications to prove the effectiveness with quantitative metrics. The main contribution seems the empirical validations.

Con:

- I think the presentation is a bit out of focus. Some sections can be left for appendix (e.g., Backpropagation in Sec 4. and Sec 5.) since some are either very standard in the literature or not really solidly experimented. Since I consider the main contribution to be the empirical evidences, the space should left for more details on those empirical experiments (for reproducibility purpose).

- the paper's idea is not particularly innovative. (yet it is not the sole reason for my scoring).

- Some relevant works on low-rank ideas have not been compared/cited. e.g.

Forrow, Aden, et al. "Statistical optimal transport via factored couplings." The 22nd International Conference on Artificial Intelligence and Statistics. PMLR, 2019.

- The writing can be improved. A lot citations/references are not particularly relevant to what this paper is about and make some parts not enough self-explained.

----
The authors addressed my concerns and I raised my evaluation ratings.

---

> ### Author Response · Authors · 2020-11-21
> **Streamlined text, factored OT, theoretical results**
>
> ## Streamlined text
> We have removed Section 5 and integrated its content into the graph transport network section, for which these improvements are an important part. We have already tested these improvements separately in Table 3.
>
> We have furthermore improved the introduction (removing many less relevant citations), pushed back the related work section so it doesn’t overwhelm the reader, and added many explanations and definitions.
>
> ## Experimental details
> We strived to include all relevant details for fully reproducing the experiments in the main paper and in Appendices H, I, and J. The source code is available in the supplementary material. If there is some missing detail, we are happy to include it.
>
> ## Contribution
> Please note that we don’t just study one particular combination of representations, but we are the first to study a sparse approximation for Sinkhorn and also the first to combine a sparse approximation with a low-rank approximation.
>
> Also, please note that in the first experiment we directly investigate the Sinkhorn approximation. Approximating the transport plan might be viewed as a ranking task, but calling this an application seems rather misleading.
>
> ## Factored OT
> We have added a comparison to factored OT (from Forrow, Aden, et al. "Statistical optimal transport via factored couplings.") to our experiments. The method performs comparatively well for high regularization, where it performs similarly to LCN-Sinkhorn (see Fig. 4, 9). However, similar to Nyström-Sinkhorn, it largely fails to approximate the transport plan at any reasonably low regularization (see Table 1, Fig. 3). Moreover, due to its iterative approach its runtime is multiple times higher than those of the other methods (see Table 9).
>
> ## Theoretical analysis
> To complement our empirical results we added a new section with a theoretical analysis. This should also improve the reader’s understanding of our method.

---

> > ### Comment · AnonReviewer4 · 2020-11-23
> > **Some additional concerns**
> >
> > Thanks for you reply. The revised paper indeed is a better version addressing some of my previous concerns. I revisited the paper and has a few additional concerns.
> >
> > 1. I realized the use of PCC (Pearson correlation coefficient) to quantify the order preserving of entries in matching plan, not for preserving the order of distances between all pairs. This is very counter-intuitive to me once I found it! Could you explain why you did not use the later choice to evaluate? The former has no application implication to me and is also not a good indicator for two plans being close.
> >
> > 2. The comparison related to factor OT is only limited to the first experiment. And in my opinion, this does not tell much about how superior the authors' approach competes empirically. Due to my first concern, the metrics used in the first experiment is only remotely connected to application values.

---

> > > ### Author Response · Authors · 2020-11-24
> > > **Resolved additional concerns**
> > >
> > > ## Ad 1) Optimal transport evaluation
> > >
> > > Since your comment can be interpreted in multiple ways we would like to address each one of them.
> > >
> > > **a) Why did we compare transport plans, not Sinkhorn distances?**
> > >
> > > We actually compare both! Table 1 also gives the relative error of the overall Sinkhorn distance and we further analyze the distance approximation in Appendix L. Note that the Sinkhorn distance is a single value and not 10^6 values like the transport plan.
> > >
> > > Appendix L also explains why we defer the Sinkhorn distance  to the appendix, instead of the transport plan: “Keep in mind that in most cases the OT plan is more important than the raw distance approximation, since it determines the training gradient and tasks like embedding alignment don’t use the distance at all. This becomes evident in the fact that sparse Sinkhorn achieves a better distance approximation than LCN-Sinkhorn but performs worse in both downstream tasks investigated in Sec. 7.”
> > >
> > > **b) Why did we analyze the transport plan, not the cost matrix C or the similarity matrix K?**
> > >
> > > Since we are interested in approximating Sinkhorn we think that analyzing its result instead of its input provides more conclusive information. Otherwise we would be missing many crucial influences.
> > >
> > > **c) Why did we use PCC to compare transport plans?**
> > >
> > > Comparing the values of 10^6 pairs is rather challenging. We primarily use the Pearson correlation coefficient (PCC) of the approximate transport plan entries $P_{ij}^{LCN}$ since this directly shows how well the values are correlated with the ground truth $P_{ij}$. In the best case this value will be 1, which means that the values lie on a perfect line with 0 width. The entries of $P_{ij}^{LCN}$ will then be exactly scaled versions of the ground truth $P_{ij}$ (around the fixed mean). In the worst case it will be 0 or even negative. We complement this with the IoU of top 0.1%, which puts more focus on the (most important) highest values.
> > >
> > > We found that PCC and IoU reflect the actual performance much better than e.g. RMSE or MAE because almost all values in the transport plan are close to 0. If a row has a single $P_{ij}$ close to 1 that means that 9,999 values will be close to 0. Measures like the RMSE and MAE then only focus on the unimportant 9,999 values close to 0, while PCC and IoU will show how well the 1 important value is reflected.
> > >
> > > To show that these summary statistics do not hide the bigger picture we also plot the raw values in Figure 1. Factored OT looks similar to the other low-rank approximation (Nyström Sinkhorn) in this comparison, i.e. it is almost flat.
> > >
> > > If you are interested nonetheless, below you can find the RMSE and MAE (std. devs are of the order 0.01e-4). Note that on average LCN-Sinkhorn performs best even for these heavily biased measures.
> > >
> > > ||MAE/10^-4 (EN-DE)|RMSE/10^-3 (EN-DE)|MAE/10^-4 (EN-ES)|RMSE/10^-3 (EN-ES)|MAE/10^-4 (EN-FR)|RMSE/10^-3 (EN-FR)|MAE/10^-4 (EN-RU)|RMSE/10^-3 (EN-RU)|
> > > |-|-|-|-|-|-|-|-|-|
> > > |Factored OT|1.58|3.94|1.63|4.83|1.62|4.70|1.49|2.87|
> > > |Multiscale OT|1.19|4.05|1.17|4.78|1.15|4.63|1.18|3.13|
> > > |Nyström Skh.|1.54|3.93|1.59|4.82|1.57|4.69|1.45|2.87|
> > > |Sparse Skh.|1.55|4.88|1.43|4.97|1.46|4.98|1.70|4.76|
> > > |LCN-Sinkhorn|1.27|3.50|1.16|3.71|1.18|3.73|1.33|3.04|
> > >
> > > ## Ad 2) End-to-end performance
> > > We are happy to hear that you agree on this point and that you appreciate the fact that we include the end-to-end performance in our evaluation. Most previous works forego this analysis.
> > >
> > > We have omitted factored OT from experiments 2 and 3 due to its OT plan approximation in the first experiment being even worse than Nyström Sinkhorn and its non-competitive runtime.
> > >
> > > If you are still curious to see the factored OT results, these are the results for experiment 2. Factored OT is more than 2x slower than LCN-Sinkhorn and other methods, which is due to factored OT’s slow iterative refinement scheme. Moreover, it completely fails to align the embeddings. While this might be expected based on experiment 1, we do realize that results as clear as this are rather uncommon. Note that we let factored OT use as many landmarks as fit into GPU memory (200, i.e. 5x more than LCN-Sinkhorn) and the partially positive results in experiment 1 show that our implementation is correct. We would also again like to highlight that our experimental setup is completely transparent and reproducible, including the source code. This result thus just highlights this task's difficulty.
> > >
> > > ||Time (s)|EN-ES|ES-EN|EN-FR|FR-EN|EN-DE|DE-EN|EN-RU|RU-EN|Avg.|
> > > |-|-|-|-|-|-|-|-|-|-|-|
> > > |Factored OT|210|0.0 ± 0.0|0.1 ± 0.0|0.2 ± 0.2|0.1 ± 0.0|0.0 ± 0.1|0.1 ± 0.0|0.0 ± 0.0|0.0 ± 0.0|0.1|
> > >
> > > ## Summary
> > > We would like to thank the reviewer for their response and are happy to have clarified any last questions. Since you agree that the paper has substantially improved since your first evaluation we would appreciate it if this would be reflected in an updated overall score.

---

### Author Response · Authors · 2020-11-22
**Overview of Improved Paper**

We would like to thank all reviewers for their helpful feedback!

We believe that we have addressed all issues in the revised version of the paper. We are now happy to have an even stronger, more approachable and thorough contribution to the community.

In particular, we have:
- Added a thorough **theoretical analysis** (Section 4, with proofs and notes in the appendix), which introduces 2 new theorems that analyze the maximum error of locally corrected Nyström for (a) uniformly distributed data and (b) clustered data. We then adapt previously established theoretical results to bound the approximation error of sparse Sinkhorn and LCN-Sinkhorn and their speed of convergence.
- Revised the first paragraph to start with a general problem description and removed less relevant work from it.
- Made it more clear in the introduction that our goal is accelerating entropy-regularized OT for point sets.
- Moved the related work section back to prevent overwhelming the reader.
- Improved the description of the Sinkhorn algorithm in and around Equation 2.
- Added Equation 3 and an improved description to better explain the approach of sparse Sinkhorn.
- Improved the motivation for locally corrected Nyström by explaining why the matrix K is full rank.
- Explicitly described how we use K_LCN.
- Improved the motivation for providing explicit gradients and moved this part to the back of the theoretical analysis.
- Streamlined the reading flow by removing the “enhanced OT” section and integrating its content into the “graph transport network” section, of which the proposed improvements (learnable unbalanced OT and multi-head OT) are an integral part, even if they are of independent interest.
- Added factored OT as another low-rank approximation baseline to the experiments.
- Improved the experimental description, e.g. better described the original Wasserstein Procrustes method.
- Better described how standard deviation is denoted in our experiments.
- Added several missing definitions, e.g. for the LSH AND-OR construction in Appendix A.

---

### Author Response · Authors · 2021-03-17
**Incorrect gradients dP/dC**

Dear readers,

We would like to highlight that the gradients proposed in Eq. 15 and the second part of Eq. 13 are incorrect. For guidance on how to derive the analytical gradient dP/dC please instead refer to the proofs in https://arxiv.org/pdf/1805.11897.pdf.

Note that the experimental results presented in this paper are not affected by this.

---

### Decision · Program_Chairs · 2021-01-07
**Final Decision**

**Decision:**

Reject

**Comment:**

The authors propose to approximate the kernel matrix used in the Sinkhorn algorithm by a combination of sparse + low rank approximation. To do so, the authors propose to compute a low rank approximation of a sparsified (thresholded below a certain value to be 0) kernel matrix using Nyström, and then correct it by adding back the true entries at non-sparse entries, after removing those obtained from the approximation. This results in a matrix whose application then results in sparse + low-rank.

The first version of the paper contained mostly experimental evidence, which was deemed a bit short by some reviewers.The authors have added theoretical material on the way. Although I believe these are worthy additions, as AC, I do not feel comfortable accepting the paper as of now, because I believe these additions were not properly reviewed. I understand this must be disappointing for the authors, who have sprinted to add new content during the rebuttal phase, but I hope they agree that the rebuttal process is not here to handle entirely new sections, but rather to improve existing parts. In particular, that section should be reviewed by authors knowledgeable on low rank kernel factorization, something I did not see in the pool of reviewers. I also believe the paper still has a few shortcomings. Taken together, I therefore recommend a re-submission.

ideas to improve the paper

- the authors claim to use Nyström on a sparsified matrix (see eq. 4). The sparsified kernel is no longer positive definite. I would like the authors to comment on this. I understand Nyström could be used naively without any psd-ness guarantees, but I think a heads-up is needed.There are, furthermore, several local/global factorizations of kernel matrices available out there (e.g. MEKA, https://www.jmlr.org/papers/v18/15-025.html), the main difference here being that the product by such approximation must be guaranteed to be positive for it to work in the Sinkhorn algorithm. I would expect that bounds in expectation to break down sometimes, and therefore result in "catastrophic" failures (i.e. nan's). I think that an algorithm that claims to improve or replace another one, and which has such blind spots, needs such additional experiments (I have read the Limitations section in Appendix B, something more precise would improve the paper). I understand these were not part of the original Nyström paper for Sinkhorn, but since this is an increment over that previous work, therefore lacking a bit its originality, more knowledge needs to be contributed.

- For instance, since the authors write an entire paragraph on this (Appendix B), I am surprised that there is not direct mention to the fact that a sparse sinkhorn may simply *not* converge, because it may not satisfy the fully indecomposable property required of matrices for Sinkhorn's algorithm to converge.

- i dont think that users have the various identities (14,15) in mind when they think about "backpropagating" through Sinkhorn. What is typically needed is to compute the differentiable properties of the regularized OT matric and/or of the regularized OT cost w.r.t. *point locations* (i.e. x_i). The statement "LCN-Sinkhorn works flawlessly with automatic backpropagation" is misleading in the sense that it ignores that problem altogether. Since so many extensions of OT today relay on that differentiability, the section, as it is written now, is problematic.

- several methods claim to be faster of more efficient than Sinkhorn to solve OT. Either these methods display faster theoretical convergence (e.g. by using acceleration) or display faster practical convergence (e.g. heavy ball variants) using synthetic, controlled datasets. Using synthetic data helps exhibit highlight relevant regimes for regularization parameters, including those where LSE Sinkhorn may converge but LCN does not work, or vice-versa. I understand that the authors' wanted to use real data, but it would be great to clarify whether that setup was used because LCN works better there (in which case this becomes more of a paper at the intersection of OT and word embeddings) or because this happened to be the first and only example the authors thought of.

---

> ### Author Response · Authors · 2021-01-15
> **Clarifications**
>
> We would like to thank the PC for the thoughtful and detailed assessment, as well as the invaluable suggestions. We look forward to revising this work for an even better and more complete contribution.
>
> To avoid future confusion we furthermore want to briefly clarify some details mentioned in the suggested improvements.
> - We do not calculate the Nyström matrix of the sparsified kernel but sparsify the Nyström matrix itself. Therefore Nyström's PSD requirements are not affected.
> - The point locations only affect the cost matrix C and we derive gradients for exactly this matrix. So the problem you describe is precisely the one we consider.

---

> > ### Comment · Area_Chair1 · 2021-03-15
> > **Question on eq. 46 in appendix**
> >
> > Dear authors,
> >
> > I appreciate your update above.
> >
> > I am not sure I understand your instantiation of the implicit function theorem in Eq. 46.
> >
> > It seems you use Eq. 46 to instantiate the implicit function theorem, and therefore somewhat state that the optimizer of regularized OT is a root of the equation C + lambda log P +1.  This implies P is basically exp( - C / lambda). However, it seems you discard all of the marginal constraints (Your \bar{P} does not even sum to 1). You do recover a much simplified (diagonal) Jacobian of P w.r.t. C, but this has no relationship to mass constrained optimal transport. Therefore I am afraid Prop. 1 seems false.

---

> > > ### Author Response · Authors · 2021-03-16
> > > **Solved via duality, Prop. 1 is still correct**
> > >
> > > Dear AC,
> > >
> > > We would like to thank you for revisiting and again becoming involved with our work. We appreciate the time you have invested and your attention to detail. Our work has substantially improved after every one of your comments.
> > >
> > > Indeed, you are partially right. Note that Prop. 1 concerns regular Sinkhorn and LCN-Sinkhorn. We thus implicitly assume that C has total support to guarantee existence of the minimizer. We should have stated this explicitly. Furthermore, the current proof indeed contains a mistake involving Eq. 46. \bar{P} lies on the boundary of the marginal constraints. The gradient thus does not have to be zero, as wrongly stated after Eq. 46. Fortunately, there is a way around this and Prop. 1 is still correct.
> > >
> > > The Sinkhorn distance is a convex optimization problem, a solution exists (assuming total support), and the positivity and marginal constraints are linear. Therefore, strong duality holds by Slater's condition. Furthermore, due to convexity the minimizer \bar{P} of the dual problem is the same as the minimizer of the primal problem. Denoting the Lagrange multipliers by α_1 to α_5 we can thus identify x=(C, α), y=P, and f(C, α, P) as the derivative of the dual w.r.t. P, i.e. f(C, α, P) = C + λ logP + λ + α_1 \vec{1}^T - α_2 \vec{1}^T + \vec{1} α_3^T - \vec{1} α_4^T - α_5; note that α_1 to α_4 are vectors, while α_5 is a matrix. At the optimum \bar{P} we have f(C, α, P) = 0 and thus g(C, α) = \bar{P}. We then obtain the same diagonal Jacobian as in Eq. 47 and the same derivative as in Eq. 48, independently of the Lagrange multipliers. The remaining proof remains the same.

---

> > > > ### Comment · Area_Chair1 · 2021-03-16
> > > > **--**
> > > >
> > > > Many thanks for answering back, despite the several limitations of this back-and-forth exercise.
> > > >
> > > > Indeed, Eq. 46 is wrong. The most important feature of the solution \bar{P} is that it has both marginals matching data, and that is not used in your current analysis.
> > > >
> > > > If you have a pointer to an updated proof, I would be interested in looking at it but I remain skeptical about the rest of your statement, and would also be curious to hear whether you have carried out numerical validation.
> > > >
> > > > The fact that \bar{P} can be written as a by product of the optimal dual variables is indeed a well known fact, P = D(s)KD(t) with your notations, where s = 𝝀 log α.
> > > >
> > > > I do not understand what you mean by
> > > >
> > > > "At the optimum \bar{P} we have f(C, α, P) = 0 and thus g(C, α) = \bar{P}."
> > > >
> > > > Assuming you are reusing the notations of your theorem B. By doing this it seems you are now moving α the same level than C. Yet α (or P) is precisely one of the outputs of the Sinkhorn optimization algorithm. Therefore it makes little sense to include it in the "solver" g. What matters is precisely differentiating α's w.r.t. C (the differential of P w.r.t. α's is easy).
> > > >
> > > > At this point I reiterate my belief that your proof looks incorrect as it stands in the draft, but that I also believe that the diagonal jacobian you mention is incorrect. Therefore I would be surprised if your other jacobians in that theorem can be validated numerically.
> > > >
> > > > An interesting reference that does such computations, putting dual variables as outputs of g,  can be seen here: https://arxiv.org/pdf/1805.11897.pdf , which could be worth adding to your bibliography.
> > > >
> > > > Since I am bit uncomfortable with this exercise, if you still claim these quantities are valid, I would encourage you to provide a link (e.g. a colab or an updated pdf) that I could examine, hoping somewhat that I am wrong.

---

> > > > > ### Author Response · Authors · 2021-03-16
> > > > > **Updated proof**
> > > > >
> > > > > We've compiled a small PDF with the updated relevant proof here: https://web.tresorit.com/l/CjDkh#IDDU8AvMEQWG3pc_6nXWfg
> > > > >
> > > > > [Edit]: Maybe it furthermore helps to remind oneself of the Wolfe dual problem when reasoning about α and P. This formulation also uses the fact that the derivative of the Lagrangian w.r.t. the primal variable (in our case P) has to be 0 at the optimum.

---

> > > > > > ### Comment · Area_Chair1 · 2021-03-16
> > > > > > **--**
> > > > > >
> > > > > > Thanks for drafting this.
> > > > > >
> > > > > > I have taken a look. I maintain this is wrong, and still not what the application of the implicit function theorem to the problem of interest here (derivation of the optimal transport matrix w.r.t the cost C) means. The Lagrange multipliers are *solved* when solving for the regularized optimal transport problem. I repeat, thet α's are not variable we wish to differentiate against, and put as arguments of g. α are implicitly defined from the marginals and C as minimizers of the regularized OT problem.
> > > > > >
> > > > > > Your Lagrangian is also needlessly heavy. Since those are equality constraints, only one multiplier with no sign constraints is needed per marginal. The lagrangian for positivity constraints is superfluous because of the logarithm.
> > > > > >
> > > > > > I invite you to look more carefully at the reference I sent you.

---

> > > > > > > ### Author Response · Authors · 2021-03-17
> > > > > > > **Apologies, Prop. 1 shortened**
> > > > > > >
> > > > > > > Indeed, you are right again. The current derivation only considers the inner optimization of P, not the optimization of the Lagrange multipliers. Interestingly, empirically this is not a horrible approximation. Nevertheless, it is wrong.
> > > > > > >
> > > > > > > The approach of deriving the gradients in the provided reference is indeed similar to ours, and can be adapted to derive the gradients we are interested in. However, the result involves computing the inverse of an NxN Hessian matrix. This would be prohibitively expensive for the large datasets we are considering. Moreover, it would not be improved by LCN-Sinkhorn. We will thus remove this part of Prop. 1 (Eq. 15 and the second part of Eq. 13) in all future versions. Unfortunately, this means that datasets with large point clouds will need to rely on regular backpropagated gradients for the transport plan P.
> > > > > > >
> > > > > > > Note that none of our experiments use gradients of P. Therefore no other part of our paper is affected by this.
> > > > > > >
> > > > > > > We would like to again thank you for your continued support and diligence. We sincerely apologize for this oversight. We will post a warning in this forum to prevent people from using the current version of the paper.

---

> > > > > > > > ### Comment · Area_Chair1 · 2021-03-29
> > > > > > > > **Happy to hear this is sorted out**
> > > > > > > >
> > > > > > > > Thanks for your answer and for your update. Indeed, it's not entirely obvious to me how these differentiability issues might be handled, even using backprop through the inverse, but maybe there's a way! Thanks anyway for sorting this out.